# Risk-sensitive learning is a winning strategy for leading an urban invasion

Alexis J Breen[1]*[†], Dominik Deffner[2,3]*

[1]Department of Human Behavior, Ecology and Culture, Max Planck Institute for Evolutionary Anthropology, Leipzig, Germany; [2]Science of Intelligence Excellence Cluster, Technical University Berlin, Berlin, Germany; [3]Center for Adaptive Rationality, Max Planck Institute for Human Development, Berlin, Germany

**\*For correspondence:**
alexis_breen@eva.mpg.de (AJB);
deffner@mpib-berlin.mpg.de (DD)

**Present address:** [†]Department of Linguistic and Cultural Evolution, Max Planck Institute for Evolutionary Anthropology, Leipzig, Germany

**Competing interest:** The authors declare that no competing interests exist.

**Abstract** In the unpredictable Anthropocene, a particularly pressing open question is how certain species invade urban environments. Sex-biased dispersal and learning arguably influence movement ecology, but their joint influence remains unexplored empirically, and might vary by space and time. We assayed reinforcement learning in wild-caught, temporarily captive core-, middle-, or edge-range great-tailed grackles—a bird species undergoing urban-tracking rapid range expansion, led by dispersing males. We show, across populations, both sexes initially perform similarly when learning stimulus-reward pairings, but, when reward contingencies reverse, male—versus female—grackles finish 'relearning' faster, making fewer choice-option switches. How do male grackles do this? Bayesian cognitive modelling revealed male grackles' choice behaviour is governed more strongly by the 'weight' of relative differences in recent foraging payoffs—i.e., they show more pronounced risk-sensitive learning. Confirming this mechanism, agent-based forward simulations of reinforcement learning—where we simulate 'birds' based on empirical estimates of our grackles' reinforcement learning—replicate our sex-difference behavioural data. Finally, evolutionary modelling revealed natural selection should favour risk-sensitive learning in hypothesised urban-like environments: stable but stochastic settings. Together, these results imply risk-sensitive learning is a winning strategy for urban-invasion leaders, underscoring the potential for life history *and* cognition to shape invasion success in human-modified environments.

## eLife assessment

This **important** study uses a multi-pronged empirical and theoretical approach to advance our understanding of animal cognition. It presents **convincing** data on how differences in learning relate to differences in the ways that male versus female animals cope with urban environments, and more generally how reversal learning may benefit animals in urban habitats.

## Introduction

Dispersal and range expansion go 'hand in hand'; movement by individuals away from a population's core is a pivotal precondition of witnessed growth in species' geographic limits (*Ronce, 2007*; *Chuang and Peterson, 2016*). Because 'who' disperses—in terms of sex—varies both within and across taxa (e.g., male-biased dispersal is dominant among fish and mammals, whereas female-biased dispersal is dominant among birds; see Table 1 in *Trochet et al., 2016*), skewed sex ratios are apt to arise at expanding range fronts, and, in turn, differentially drive invasion dynamics (*Miller et al., 2011*). Female-biased dispersal, for instance, can 'speed up' staged invertebrate invasions by increasing offspring production (*Miller and Inouye, 2013*). Alongside sex-biased dispersal, cognition is also argued to contribute to species' colonisation capacity, as novel environments inevitably

present novel (foraging, predation, shelter, and social) challenges that newcomers need to surmount in order to settle successfully (*Wright et al., 2010*; *Sutter and Kawecki, 2009*). Indeed, a growing number of studies show support for this supposition, at least for those animals thriving in urban environments (recent reviews: *Lee and Thornton, 2021*; *Szabo et al., 2020*; *Barrett et al., 2019*; but see: *Vincze and Kovács, 2022*). Carefully controlled choice tests show, for example, urban-dwelling individuals will learn novel stimulus-reward pairings more readily than do rural-dwelling counterparts (*Batabyal and Thaker, 2019*). Similarly, urban-dwellers will more frequently figure out how to overcome experimenter-placed obstacles blocking known food resources, compared to conspecific rural-dwellers (*Mazza and Guenther, 2021*). And how often such hidden food rewards are successfully accessed positively relates to how long specific subspecies have lived commensually with humans in urban environments (*Vrbanec et al., 2021*). Taken together, these data in both resident urban and non-urban species as well as urban invasive species generally support the view that urban environments favour particular learning phenotypes. It is perhaps surprising, then, that despite their apparent independent influence on movement ecology, the potential interactive influence of sex-biased dispersal and learning on successful urban invasion remains unexamined empirically (but for related theoretical study, see: *Liedtke and Fromhage, 2021b*; *Liedtke and Fromhage, 2021a*), especially as the dynamics of urban environments appear distinctly demanding.

Urban environments are hypothesised to be both stable and stochastic: more specifically, urbanisation is thought to lead to stabilisation in some aspects of biotic structure, including predation pressure, thermal habitat, and resource availability, and to enhanced abiotic disruption, such as anthropogenic noise and light pollution (reviewed in: *Shochat et al., 2006*; *Francis and Barber, 2013*; *Gaston et al., 2013*). Seasonal survey data from (sub)urban British neighborhoods show, for example, 40–75% of households provide supplemental feeding resources for birds (e.g., seed, bread, and peanuts; *Cowie and Hinsley, 1988*; *Davies et al., 2009*), the density of which can positively relate to avian abundance within an urban area (*Fuller et al., 2008*). But such supplemental feeding opportunities are theorised to be traded off against unpredictable and likely fitness-affecting anthropogenic disturbances (e.g., automobile and airplane traffic; as outlined in *Frid and Dill, 2002*). Experimental data show, for example, the more variable are traffic noise and pedestrian presence, the more negative are such human-driven effects on birds' sleep (*Grunst et al., 2021*), mating (*Blickley et al., 2012*), and foraging behaviour (*Fernández-Juricic, 2000*). Understanding how and why particular species successfully inhabit seemingly stable but stochastic urban environments remains an open and timely question, as highlighted by recent concerns over (in)vertebrates' coping ability to current and projected levels of urbanisation (*Eisenhauer and Hines, 2021*; *Li et al., 2022*).

Great-tailed grackles (*Quiscalus mexicanus*; henceforth, grackles) are an excellent model for empirical examination of the interplay between sex-biased dispersal, learning, and ongoing urban-targeted rapid range expansion: over the past ~150 years, they have seemingly shifted their historically human-associated niche to include more variable urban environments (e.g., arid habitat; *Summers et al., 2023*; *Haemig, 1978*), moving from their native range in Central America into much of the United States, with several first-sightings spanning as far north as Canada (*Dinsmore and Dinsmore, 1993*; *Wehtje, 2003*; *Fink et al., 2020*). Notably, the record of this urban invasion is heavily peppered with first sightings involving a single or multiple male(s) (41 of 63 recorded cases spanning most of the twentieth century; *Dinsmore and Dinsmore, 1993*). Moreover, recent genetic data show, when comparing grackles within a population, average relatedness: (i) is higher among females than among males; and (ii) decreases with increasing geographic distance among females; but (iii) is unrelated to geographic distance among males; hence, confirming urban invasion in grackles is male-led via sex-biased dispersal (*Sevchik et al., 2022*). Considering these life history and genetic data in conjunction with data on grackle wildlife management efforts (e.g., pesticides, pyrotechnics, and sonic booms; *Luscier, 2018*), it seems plausible that—regardless of their shared human and urban 'history'—urban invasion might drive differential learning between male and female grackles, potentially resulting in a spatial sorting of the magnitude of this sex difference with respect to population establishment age (i.e., sex-effect: newer population > older population; *Phillips et al., 2010*). Indeed, irrespective of population membership, such sex differences could come about via differential reliance on learning strategies mediated by an interaction between grackles' polygynous mating system and male-biased dispersal system (see our in-depth discussion on this point). Population membership might, in turn, differentially moderate the magnitude of any such sex-effect since an edge population, even though

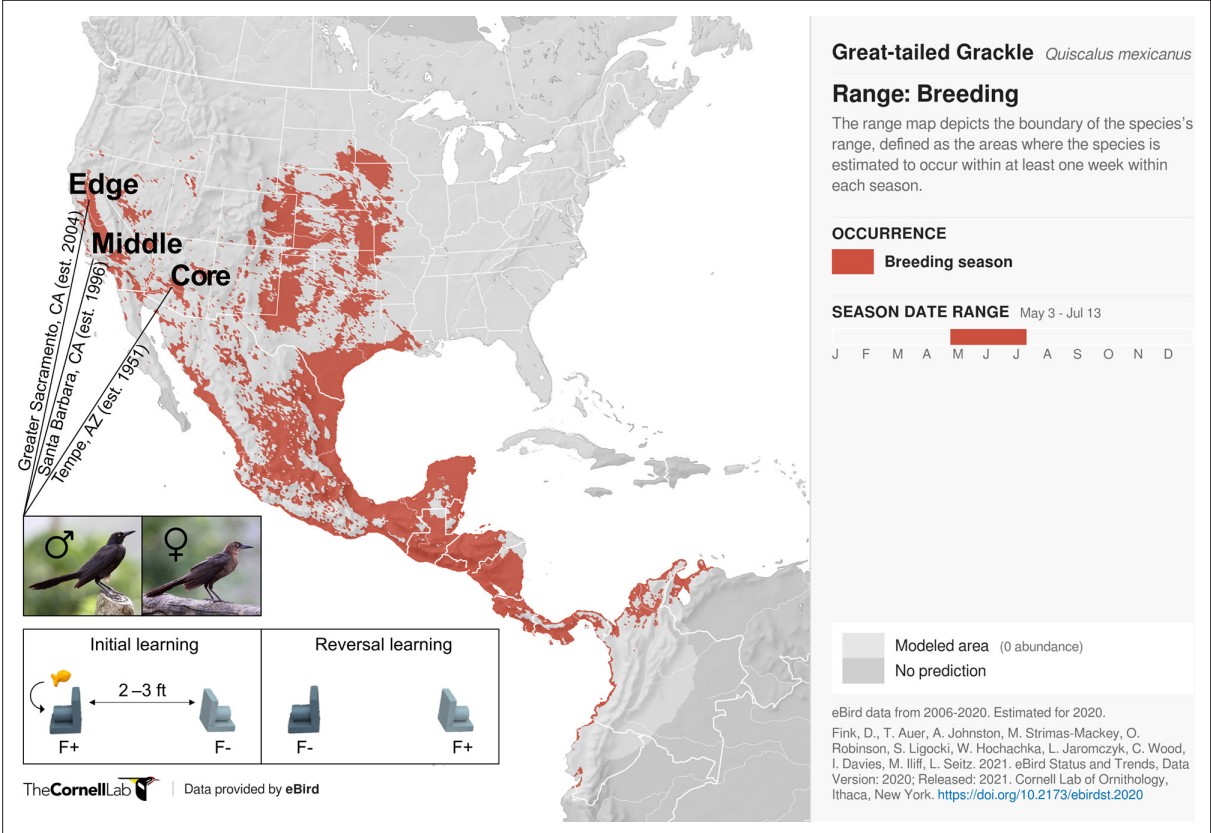

**Figure 1.** Participants and experimental protocol. Thirty-two male and 17 female wild-caught, temporarily captive great-tailed grackles either inhabiting a core (17 males, 5 females), middle (4 males, 4 females), or edge (11 males, 8 females) population of their North American breeding range (establishment year: 1951, 1996, and 2004, respectively) are participants in the current study (grackle images: Wikimedia Commons). Each grackle is individually tested on a two-phase reinforcement learning paradigm: *initial learning*, two colour-distinct tubes are presented, but only one coloured tube (e.g., dark grey) contains a food reward (F+ versus F-); *reversal learning*, the stimulus-reward tube pairings are swapped. The learning criterion is identical in both learning phases: 17 F+ choices out of the last 20 choices, with trial 17 being the earliest a grackle can successfully finish (for details, see Materials and methods).

urban, could still pose novel challenges—e.g., by requiring grackles to learn novel daily temporal foraging patterns such as when and where garbage is collected (grackles appear to track this food resource: *Rodrigo et al., 2021*).

Here, we test the hypothesis that sex differences in dispersal are related to sex differences in learning in an urban invader—grackles. Specifically, we examine whether, and, if so, how sex mediates learning across 32 male and 17 female wild-caught, temporarily captive grackles either inhabiting a core (17 males, 5 females), middle (4 males, 4 females), or edge (11 males, 8 females) population of their North American range (based on year-since-first-breeding: 1951, 1996, and 2004, respectively; details in Materials and methods; *Figure 1*). To do this, we collated, cleaned, and curated existing reinforcement learning data (see Data provenance), wherein novel stimulus-reward pairings are presented (i.e., *initial learning*), and, once successfully learned via reaching a criterion, these reward contingencies are reversed (i.e., *reversal learning*). As range expansion should disfavour slow, error-prone learning strategies, we expect male and female grackles to differ across at least two reinforcement learning behaviours: speed (trials to criterion) and choice-option switches (times alternating between available stimuli). Specifically, as documented in our preregistration (see *Supplementary file 1*), if learning and dispersal relate we expect male—versus female—grackles: (predictions 1 and 2) to be faster to, firstly, learn a novel colour-reward pairing, and secondly, reverse their colour preference when the colour-reward pairing is swapped; and (prediction 3) to make fewer choice-option switches during their colour-reward learning. Finally, we further expect (prediction 4) such sex-mediated differences in learning to be more pronounced in grackles living at the edge, rather than the intermediate and/or core region of their range.

To comprehensively examine links between sex-biased dispersal and learning in urban-invading grackles, we employ a combination of Bayesian computational and cognitive modelling methods, and both agent-based and evolutionary simulation techniques. Specifically, our paper proceeds as follows: (i) we begin by describing grackles' reinforcement learning and testing our predictions using multi-level Bayesian Poisson models; (ii) we next 'unblackbox' candidate learning mechanisms generating grackles' reinforcement learning using a multi-level Bayesian reinforcement learning model, as ultimately the mechanisms of behaviour should be the target of selection; (iii) we then try to replicate our behavioural data via agent-based forward simulations, to determine if our inferred learning mechanisms sufficiently explain our grackles' reinforcement learning; and (iv) we conclude by examining the evolutionary implications of variation in these learning mechanisms in hypothesised urban-like (or not) environments—i.e., statistical settings that vary in both stability and stochasticity—via algorithmic optimisation.

## Results

### Reinforcement learning behaviour

We do not observe credible population-level differences in grackles' reinforcement learning (*Supplementary file 2a and b*). As such, we compare male and female grackles' reinforcement learning across populations. Both sexes start out as similar learners, finishing initial learning in comparable trial numbers (mean trials-to-finish: males, 35; females, 36; *Figure 2A* and *Supplementary file 2a*), and with comparable counts of choice-option switches (mean switches-at-finish: males, 14; females, 14; *Figure 2* and *Supplementary file 2b*). Indeed, the male-female (M-F) posterior contrasts for both behaviours centre around zero, evidencing no sex-effect (*Figure 2C*). Once reward contingencies reverse, however, male—versus female—grackles finish this 'relearning' faster by taking fewer trials (mean trials-to-finish: males, 67; females, 81; *Figure 2B* and *Supplementary file 2a*), and by making fewer choice-option switches (mean switches-at-finish: males, 25; females, 36; *Figure 2B* and *Supplementary file 2b*). The M-F posterior contrasts, which lie almost entirely below zero, clearly capture this sex-effect (*Figure 2F* and *Supplementary file 2a and b*). Environmental unpredictability, then, dependably directs disparate reinforcement learning trajectories between male and female grackles, supporting our overall expectation of sex-mediated differential learning in urban-invading grackles.

### Reinforcement learning mechanisms

Because (dis)similar behaviour can result from multiple latent processes (*McElreath, 2018*), we next employ computational methods to delimit reinforcement learning mechanisms. Specifically, we adapt a standard multi-level Bayesian reinforcement learning model (from *Deffner et al., 2020*; but see also: *Farrell and Lewandowsky, 2018*; *McElreath et al., 2005*; *Sutton and Barto, 2018*), which we validate a priori via agent-based simulation (see Materials and methods and *Supplementary file 1*), to estimate the contribution of two core latent learning parameters to grackles' reinforcement learning: the *information-updating rate* $\varphi$ (How rapidly do learners 'revise' knowledge?) and the *risk-sensitivity rate* $\lambda$ (How strongly do learners 'weight' knowledge?). Both learning parameters capture individual-level internal response to incurred reward-payoffs, but they differentially direct any reward sensitivity back on choice behaviour due to their distinct definitions (full mathematical details in Materials and methods). For $\varphi$, stronger reward sensitivity (bigger values) means faster internal updating about stimulus-reward pairings, which translates behaviourally into faster learning about 'what to choose'. For $\lambda$, stronger reward sensitivity (bigger values) means stronger internal determinism about seeking the non-risk foraging option (i.e., the one with the higher expected payoffs based on prior experience), which translates behaviourally into less choice-option switching i.e., 'playing it safe'. In sum, conditional on our reinforcement learning model, we reverse engineer which values of our learning parameters most likely produce grackles' choice behaviour—an analytical advantage over less mechanistic methods (*McElreath, 2018*).

Looking at our reinforcement learning model's estimates between populations to determine replicability, we observe, in initial learning, the information-updating rate $\varphi$ of core- and edge-inhabiting male grackles is largely lower than that of female counterparts (M-F posterior contrasts lie more below zero; *Figure 2G* and *Supplementary file 2c*), with smaller sample size likely explaining the middle population's more uncertain estimates (M-F posterior contrasts centre widely around zero; *Figure 2G*

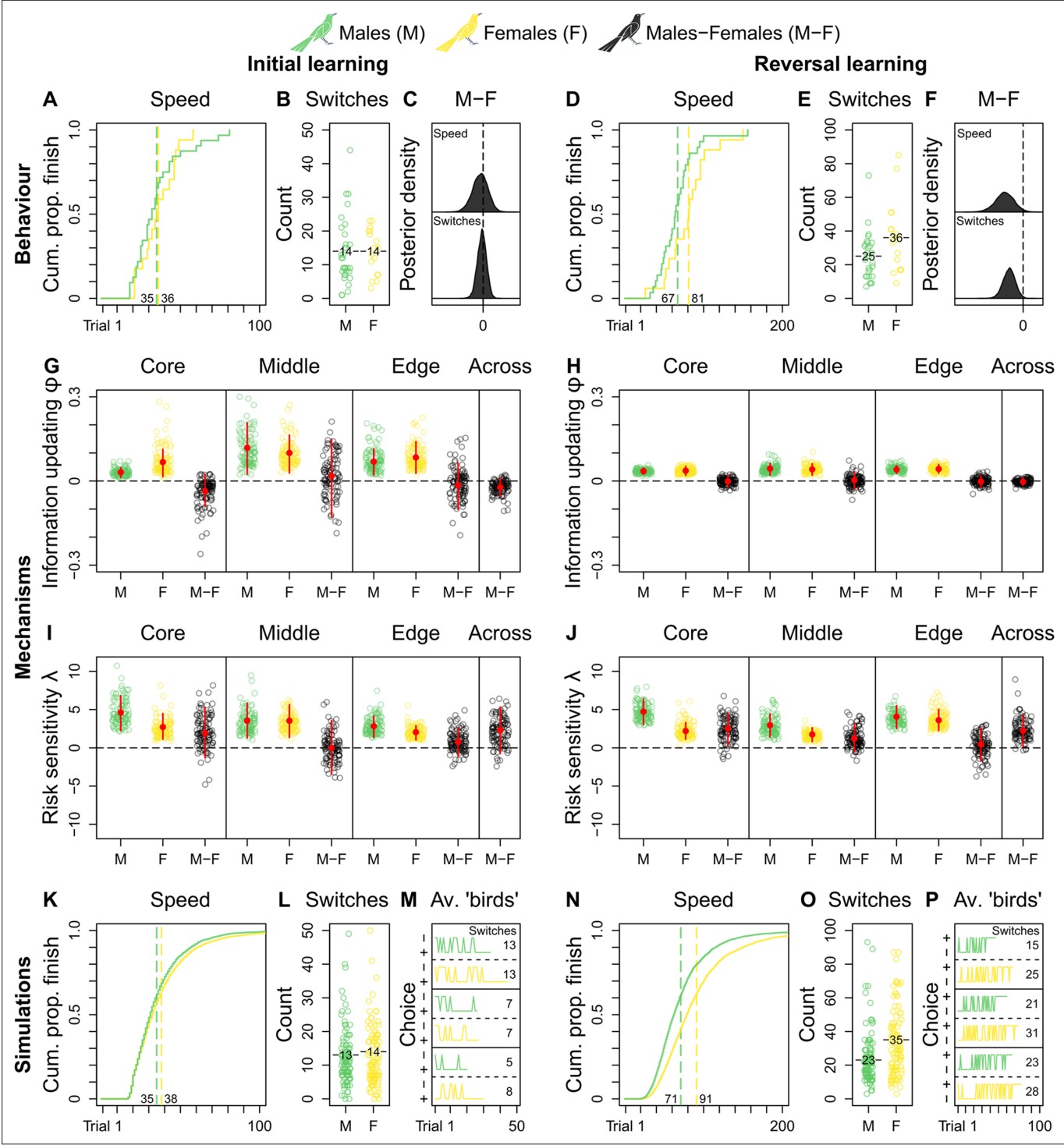

**Figure 2.** Grackle reinforcement learning. *Behaviour.* Across-population reinforcement learning speed and choice-option switches in (**A–B**) initial learning (M, 32; F, 17) and (**D–E**) reversal learning (M, 29; F, 17), with (**C, F**) respective posterior estimates and male-female (M-F) contrasts. *Mechanisms.* Within- and across-population estimates and contrasts of *information-updating rate φ* and *risk-sensitivity rate λ* in (**G, I**) initial learning and (**H, J**) reversal learning. In (**G–J**) open circles show 100 random posterior draws; red filled circles and red vertical lines show posterior means and 89% highest posterior density intervals, respectively. *Simulations.* Across-population forward simulated reinforcement learning speed and choice-option switches in (**K–M**) initial learning and (**N–P**) reversal learning. In (**K, N**) the full simulation sample is plotted; in (**L, O**) open circles show 100 random simulant draws; and (**M, P**) show three random 'average' M or F simulants. Note (**K, N**) x-axes are cut to match (**A, D**) x-axes. Means are plotted/labelled in (**A, B, D, E, K, L, N, O**). Plots (**G–P**) are generated via model estimates using our full sample size (M, 32; F, 17).

*Figure 2 continued on next page*

*Figure 2 continued*

The online version of this article includes the following figure supplement(s) for figure 2:

**Figure supplement 1.** Extra initial learning trials do not noticeably influence grackles' reversal learning.

and *Supplementary file 2c*); while in reversal learning, the information-updating rate $\varphi$ of both sexes is nearly identical irrespective of population membership, with females dropping to the reduced level of males (M-F posterior contrasts centre closely around zero; *Figure 2H* and *Supplementary file 2c*). Therefore, the information-updating rate $\varphi$ across male and female grackles is initially different (males < females), but converges downwards over reinforcement learning phases (across-population M-F posterior contrasts first lie mostly below, and then, tightly bound zero; *Figure 2G and H* and *Supplementary file 2c*).

These primary mechanistic findings are, at first glance, perplexing: if male grackles generally outperform female grackles in reversal learning (*Figure 2D–F*), why do all grackles ultimately update information at matched, dampened pace? This apparent conundrum, however, in fact highlights the potential for multiple latent processes to direct behaviour. Case in point: the risk-sensitivity rate $\lambda$ is distinctly higher in male grackles, compared to female counterparts, regardless of population membership and learning phase (M-F posterior contrasts lie more, if not mostly, above zero; *Figure 2I and J* and *Supplementary file 2d*), outwith the middle population in initial learning likely due to sample size (M-F posterior contrasts centre broadly around zero; *Figure 2I* and *Supplementary file 2d*). In other words, choice behaviour in male grackles is more strongly governed by relative differences in predicted reward-payoffs, as spotlighted by across-population M-F posterior contrasts that lie mostly above zero in initial learning, and entirely above zero in reversal learning (*Figure 2I and J* and *Supplementary file 2d*). Thus, these combined mechanistic data reveal, when reward contingencies reverse, male—versus female—grackles 'relearn' faster via pronounced risk-sensitive learning.

## Agent-based forward simulations and replication of reinforcement learning

To determine definitively whether our estimated learning parameters are sufficient to generate grackles' observed reinforcement learning, we conduct agent-based forward simulations. Agent-based forward simulations are posterior predictions, and they provide insight into the implied model dynamics and overall usefulness of our reinforcement learning model. Specifically, agent-based forward simulations allow us to ask—what would a 'new' grackle 'do', given our reinforcement learning model parameter estimates? It is important to ask this question because, in parameterising our model, we may have overlooked a critical contributing mechanism to grackles' reinforcement learning. Such an omission is invisible in the raw parameter estimates; it is only betrayed by the estimates in actu. The simulation thus assigns 'birds' random information-updating rate $\varphi$ and risk-sensitivity rate $\lambda$ draws (whilst maintaining their correlation structure), and tracks their reinforcement learning. By comparing these synthetic data to our real data, we gain valuable insight into the learning and choice behaviour implied by our reinforcement learning model results. If no critical contributing mechanism(s) went overlooked, simulated 'birds' should behave similar to our real birds. A disparate mapping between simulated 'birds' and our real birds, however, would mean more work is needed with respect to model parameterisation that captures the causal, mechanistic dynamics behind real birds' reinforcement learning (for an example of this happening in the human reinforcement learning literature, see *Deffner et al., 2020*).

Ten thousand synthetic reinforcement learning trajectories (5000 'males' and 5000 'females'), together, compellingly show our 'birds' behave just like our grackles: 'males' outpace 'females' in reversal but not in initial learning (mean trials-to-finish in initial and reversal learning: 'males', 35 and 71; 'females', 38 and 91; respectively; *Figure 2K and N*); and 'males' make fewer choice-option switches in initial but not in reversal learning, compared to 'females' (mean switches-at-finish in initial and reversal learning: 'males', 13 and 23; 'females', 14 and 35; respectively; *Figure 2L and O*). *Figure 2M and P* show, respectively, synthetic initial and reversal learning trajectories by three average 'males' and three average 'females' (i.e., simulants informed via learning parameter estimates that average over our posterior distribution), for the reader interested in representative individual-level reinforcement

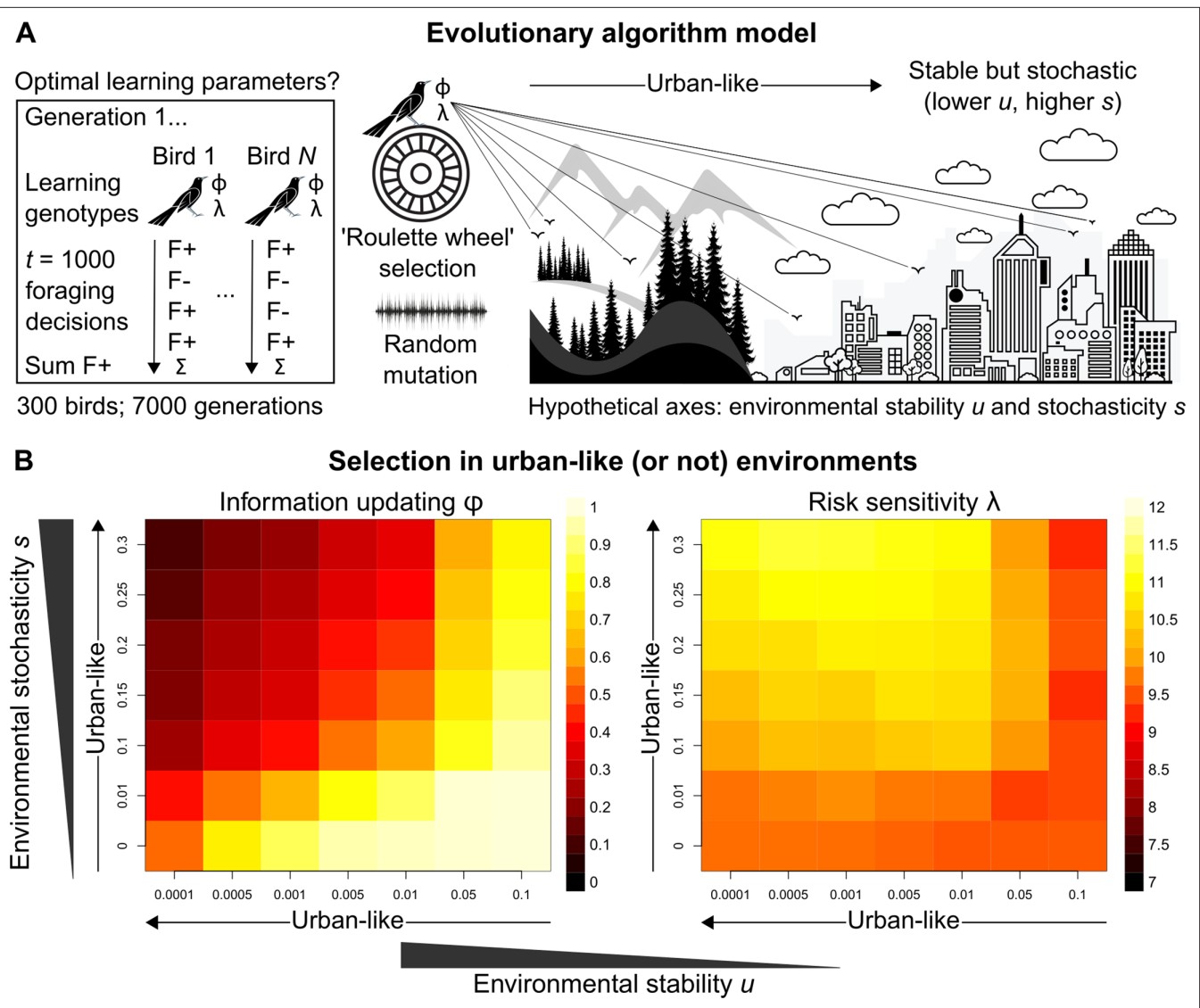

**Figure 3.** Evolutionary optimality of strategising risk-sensitive learning. (**A**) Illustration of our evolutionary algorithm model to estimate optimal learning parameters that evolve under systematically varied pairings of two hypothesised urban ecology axes: *environmental stability **u*** and *environmental stochasticity **s***. Specifically, 300-member populations run for 10 independent 7000-generation simulations per pairing, using 'roulette wheel' selection (parents are chosen for reproduction with a probability proportional to collected F+ rewards out of 1000 choices) and random mutation (offspring inherit learning genotypes with a small deviation in random direction). (**B**) Mean optimal learning parameter values discovered by our evolutionary model (averaged over the last 5000 generations). As the statistical environment becomes seemingly more urban-like (lower ***u*** and higher ***s*** values), selection should favour lower information-updating rate ***φ*** and higher risk-sensitivity rate ***λ*** (darker and lighter squares in left and right plot, respectively). We note arrows are intended as illustrative aids and do not correspond to a linear scale of 'urbanness'.

learning dynamics. Such quantitative replication confirms our reinforcement learning model results sufficiently explain our behavioural sex-difference data.

### Selection of reinforcement learning mechanisms under urban-like environments

Learning mechanisms in grackles obviously did not evolve to be successful in the current study; instead, they likely reflect selection pressures and/or adaptive phenotypic plasticity imposed by urban environments (*Blackburn et al., 2009*; *Sol et al., 2013*; *Lee and Thornton, 2021*; *Vinton et al., 2022*; *Caspi et al., 2022*). Applying an evolutionary algorithm model (*Figure 3A*), we conclude by examining how urban environments might favour different information-updating rate *φ* and risk-sensitivity rate *λ*

values, by estimating optimal learning strategies in settings that differ along the two ecological axes thought to represent urban environmental characteristics (see Introduction): *environmental stability u* (How often does optimal behaviour change?) and *environmental stochasticity s* (How often does optimal behaviour fail to payoff?). In reality, much additional (a)biotic and life history information is relevant to thoroughly testing such urban eco-evolutionary theory. But our evolutionary model is a necessary and useful first step towards addressing targeted research questions about the interplay between learning phenotype and apparent environmental characteristics.

Strikingly, under hypothesised urban-like (i.e., stable but stochastic) environments, our evolutionary model shows the learning parameter constellation robustly exhibited by males grackles in our study—low information-updating rate $\varphi$ and high risk-sensitivity rate $\lambda$—should be favoured by natural selection (darker and lighter squares in, respectively, the left and right plots in *Figure 3B*). These results imply, in seemingly urban and other statistically similar environments, learners benefit by averaging over prior experience (i.e., gradually updating 'beliefs'), and by informing behaviour based on this experiential history (i.e., proceeding with 'caution'), highlighting the adaptive value of strategising risk-sensitive learning in such settings. We note that, for most rows of environmental stochasticity $s$, intermediate levels of environmental stability $u$ generated the highest risk-sensitivity rate $\lambda$ values, while risk-sensitivity rate $\lambda$ values were slightly lower for both extremes of environmental stability $u$. The most plausible explanation for this pattern is that when the environment changes at intermediate levels, it is advantageous for learners to be fairly risk-sensitive (like our grackles), so that behaviour can be both effective and effectively adjusted. By contrast, in environments with either minimal or maximal environmental change, selection should be weaker for, or favouring lower degrees of, risk-sensitive learning because optimal behaviour is either more-or-less constant or constantly changing, respectively.

## Discussion

Mapping a full pathway from behaviour to mechanisms through to selection and adaptation, we show risk-sensitive learning is a viable strategy to help explain how male grackles—the dispersing sex—currently lead their species' remarkable North American urban invasion. Specifically, in wild-caught, temporarily captive core-, middle-, or edge-range grackles, we show: (i) irrespective of population membership, male grackles outperform female counterparts on stimulus-reward reversal reinforcement learning, finishing faster and making fewer choice-option switches; (ii) they achieve their superior reversal learning performance via pronounced risk-sensitive learning (low $\varphi$ and high $\lambda$), as 'unblackboxed' by our mechanistic model; (iii) these learning mechanisms sufficiently explain our sex-difference behavioural data: because we replicate our results using agent-based forward simulations; and (iv) risk-sensitive learning—i.e., low $\varphi$ and high $\lambda$—appears advantageous in hypothesised urban-like environments (stable but stochastic settings), according to our evolutionary model. These results set the scene for future comparative research.

The term 'behavioural flexibility'—broadly defined as some 'attribute', 'cognition', 'characteristic', 'feature', 'trait', and/or 'quality' that enables animals to adapt behaviour to changing circumstances (*Coppens et al., 2010*; *Audet and Lefebvre, 2017*; *Barrett et al., 2019*; *Lea et al., 2020*)—has previously been hypothesised to explain invasion success (*Wright et al., 2010*), including that of grackles (*Summers et al., 2023*). But as eloquently argued elsewhere (*Audet and Lefebvre, 2017*), this term is conceptually uninformative, given the many ways in which it is applied and assessed. Of these approaches, reversal learning and serial—multiple back-to-back—reversal learning tasks are the most common experimental assays of behavioural flexibility (non-exhaustive examples of each assay in: bees; *Strang and Sherry, 2014*; *Raine and Chittka, 2012*; birds; *Bond et al., 2007*; *Morand-Ferron et al., 2022*; fish; *Lucon-Xiccato and Bisazza, 2014*; *Bensky and Bell, 2020*; frogs; *Liu et al., 2016*; *Burmeister, 2022*; reptiles; *Batabyal and Thaker, 2019*; *Gaalema, 2011*; primates; *Cantwell et al., 2022*; *Lacreuse et al., 2018*; rodents; *Rochais et al., 2021*; *Boulougouris et al., 2007*). We have shown, however, at least for our grackles, faster reversal learning is governed primarily by risk-sensitive learning, so: firstly, these go-to reversal learning assays do not necessarily measure the unit they claim to measure (a point similarly highlighted in: *Aljadeff and Lotem, 2021*; *Federspiel et al., 2017*); and secondly, formal models based on the false premise that variation in learning speed relates to variation in behavioural flexibility require reassessment (*Lea et al., 2020*; *Blaisdell et al., 2021*; *Logan et al., 2022*; *Lukas et al., 2024*; *Logan et al., 2023a*; *Logan et al., 2023b*). Indeed,

perhaps unsurprisingly (*Figure 3B*), grackles that learn how to learn in serial reversal experiments (i.e., high $u$ and low $s$ environments) do so primarily through shifts up in information-updating rates (*Lukas et al., 2024*), though this consecutive relearning task does not resemble the foraging scenarios or the apparent environmental dynamics grackles navigate day-to-day in urban environments (*Fronimos et al., 2011*; *Rodrigo et al., 2021*; *Shochat et al., 2006*; *Francis and Barber, 2013*; *Gaston et al., 2013*). Heeding previous calls (*Dukas, 1998*; *McNamara and Houston, 2009*; *Fawcett et al., 2013*), our study provides an analytical solution to facilitate productive research on proximate and ultimate explanations of seemingly flexible (or not) behaviour: because we publicly provide step-by-step code to examine individual decision-making, two core underlying learning mechanisms, and their theoretical selection and benefit (see GitHub, copy archived at *Breen, 2024*), which can be tailored to specific research questions. The reinforcement learning model, for example, generalises to, in theory, a variety of choice-option paradigms (*Barrett, 2023*), and these learning models can be extended to estimate asocial *and* social influence on individual decision-making (e.g., *McElreath et al., 2005*; *Aplin et al., 2017*; *Barrett et al., 2017*; *Deffner et al., 2020*; *Chimento et al., 2022*), facilitating insight into the multi-faceted feedback process between individual cognition and social systems (*Tump et al., 2024*). Our open-access analytical resource thus allows researchers to dispense with the umbrella term behavioural flexibility, and to biologically inform and interpret their science— only then can we begin to meaningfully examine the functional basis of behavioural variation across taxa and/or contexts.

## Ideas and speculation

Related to this final point, it is useful to outline how additional drivers outwith sex-biased risk-sensitive learning might contribute towards urban-invasion success in grackles, too. Grackles exhibit a polygynous mating system, with territorial males attracting multiple female nesters (*Johnson et al., 2000*). Recent learning 'style' simulations show the sex with high reproductive skew approaches pure individual learning, while the other sex approaches pure social learning (*Smolla et al., 2019*). During population establishment, then, later-arriving female grackles could rely heavily on vetted information provided by male grackles on 'what to do' (*Wright et al., 2010*), as both sexes ultimately face the same urban-related challenges. Moreover, risk-sensitive learning in male grackles should help reduce the elevated risk associated with any skew towards acquiring knowledge through individual learning. And as the dispersing sex this process would operate independently of their proximity to a range front—a pattern suggestively supported by our mechanistic data (i.e., risk-sensitivity: males > females; *Figure 2G and H*). As such, future research on potential sex differences in social learning propensity in grackles seem particularly prudent, alongside systematic surveying of population-level environmental and fitness components across spatially (dis)similar populations.

The lack of spatial replicates in the existing data set used herein inherently poses limitations on inference. Nevertheless, the currently available data do not show meaningful population-level behavioural or mechanistic differences in grackles' reinforcement learning, and we should thus be cautious about speculating on between-population variation. But it is worth noting that phenotypic filtering by invasion stage is not a compulsory signature of successful (urban) invasion. Instead, phenotypic plasticity and/or inherent species trait(s) may be facilitators (*Blackburn et al., 2009*; *Sol et al., 2013*; *Lee and Thornton, 2021*; *Vinton et al., 2022*; *Caspi et al., 2022*). For urban-invading grackles, both of these biological explanations seem strongly plausible, given: firstly, grackles' highly plastic foraging and nesting ecology (*Selander and Giller, 1961*; *Davis and Arnold, 1972*; *Wehtje, 2003*); secondly, grackles' apparent historic and current realised niche being—albeit in present day more variable—urban and human-associated environments, a consistent habit preference that cannot be explained by changes in habitat availability or habitat connectivity (*Summers et al., 2023*); and finally, our combined behavioural, mechanistic, and evolutionary modelling results showing environments apparently approaching grackles' general species niche—human and urban environments—select for particular traits that exist across grackle populations (here, sex-biased risk-sensitive learning). Admittedly, our evolutionary model is not a complete representation of urban ecology dynamics. Relevant factors—e.g., spatial dynamics and realistic life histories—are missing. These omissions are tactical ones. Our evolutionary model solely focuses on the response of reinforcement learning parameters to hypothesised urban-like (or not) environmental statistics, providing a baseline for future studies to build on; for example, it would be interesting to investigate such selection on learning parameters of

'true' invaders and not their descendants, a logistically tricky but nonetheless feasible research possibility (e.g., *Duckworth and Badyaev, 2007*).

## Conclusions

By revealing across-population interactive links between the dispersing sex and risk-sensitive learning in an urban invader (grackles), these analytically replicable insights, coupled with our finding that hypothesised urban-like environments favour pronounced risk-sensitivity, imply risk-sensitive learning is a winning strategy for urban-invasion leaders. Our modelling methods, which we document in-depth and make freely available, can now be comparatively applied, establishing a biologically meaningful analytical approach for much-needed study on (shared or divergent) drivers of geographic and phenotypic distributions (*Somveille et al., 2018*; *Bro-Jørgensen et al., 2019*; *Lee and Thornton, 2021*; *Breen et al., 2021*; *Breen, 2021*; *Deffner et al., 2022*).

## Materials and methods

### Data provenance

The current study uses data from two types of sources: publicly archived data at the Knowledge Network for Biocomplexity (*Logan, 2018*; *Logan et al., 2021*); or privately permissed access to AJB of (at the time) unpublished data by Grackle Project principal investigator Corina Logan, who declined participation on this study. We note these shared data are now also available at the Knowledge Network for Biocomplexity (*Logan et al., 2023c*). Finally, we note the cleaned versions of at least part of these data (see Reinforcement learning criterion) are available at our GitHub repository.

### Data contents

The data used herein chart colour-reward reinforcement learning performance from 32 male and 17 female wild-caught, temporarily captive grackles inhabiting one of three study sites that differ in their range expansion demographics i.e., defined as a core, middle, or edge population (based on time-since-settlement population growth dynamics, as outlined in *Chuang and Peterson, 2016*). Specifically: (i) Tempe, Arizona (17 males and 5 females)—herein, the core population (estimated to be breeding since 1951, by adding the average time between first sighting and first breeding to the year first sighted; *Wehtje, 2003*; *Wehtje, 2004*); (ii) Santa Barbara, California (4 males and 4 females)—herein, the middle population (known to be breeding since 1996; *Lehman, 2020*); and (iii) Greater Sacramento, California (11 males and 8 females)—herein, the edge population (known to be breeding since 2004; *Hampton, 2004*).

### Experimental protocol

A step-by-step description of the experimental protocol carried out by the original experimenters is reported elsewhere (*Blaisdell et al., 2021*). As such, below we detail only the protocol for the colour-reward reinforcement learning test that we analysed herein.

### Reinforcement learning test

The reinforcement learning test consists of two experimental phases (*Figure 1*): (i) stimulus-reward initial learning and (ii) stimulus-reward reversal learning. In both experimental phases, two different coloured tubes are used: for Santa Barbara grackles, gold and grey; for all other grackles, light and dark grey. Each tube consists of an outer and inner diameter of 26 mm and 19 mm, respectively; and each is mounted to two pieces of plywood attached at a right angle (entire apparatus: 50 mm wide×50 mm tall×67 mm deep); thus resulting in only one end of each coloured tube being accessible (*Figure 1*).

In initial learning, grackles are required to learn that only one of the two coloured tubes contains a food reward (e.g., dark grey); this colour-reward pairing is counterbalanced across grackles within each study site. Specifically, the rewarded and unrewarded coloured tubes are placed—either on a table or on the floor—in the centre of the aviary run (distance apart: table, 2 feet; floor, 3 feet), with the open tube-ends facing, and perpendicular to, their respective aviary side wall. Which coloured tube is placed on which side of the aviary run (left or right) is pseudorandomised across trials. A trial begins at tube placement, and ends when a grackle has either made a tube choice or the maximum

trial time has elapsed (8 min). A tube choice is defined as a grackle bending down to examine the contents (or lack thereof) of a tube. If the chosen tube contains food, the grackle is allowed to retrieve and eat the food, before both tubes are removed and the rewarded coloured tube is rebaited out of sight (for the grackle). If a chosen tube does not contain food, both tubes are immediately removed. Each grackle is given, first, up to 3 min to make a tube choice, after which a piece of food is placed equidistant between the tubes to entice participation; and then, if no choice has been made, an additional 5 min maximum, before both tubes are removed. All trials are recorded as either correct (choosing the rewarded coloured tube), incorrect (choosing the unrewarded coloured tube), or incomplete (no choice made). To successfully finish initial learning, a grackle must meet the learning criterion, detailed below.

In reversal learning, grackles are required to learn that the colour-reward pairing has been swapped; that is, the previously unrewarded coloured tube (e.g., light grey) now contains a food reward (*Figure 1*). The protocol for this second and final experimental phase is identical to that, described above, of initial learning.

## Reinforcement learning criterion

For all grackles in the current study, we apply the following learning criterion: to successfully finish their respective learning phase, grackles must make a correct choice in 17 of the most recent 20 trials. Therefore, the earliest a grackle can successfully finish initial or reversal learning in the current study is at trail 17. This applied learning criterion is the most compatible with respect to previous learning criteria used by the original experimenters. Specifically, *Logan, 2018*, and *Logan et al., 2021*, used a fixed-window learning criterion for core- and middle-population grackles, in which grackles were required to make 17 out of the last 20 choices correctly, with a minimum of eight and nine correct choices across the last two sets of 10 trials, assessed at the end of each set. If a core- or middle-population grackle successfully satisfied the fixed-window learning criterion, the grackle was assigned by Logan or colleagues the final trial number for that set (e.g., 20, 30, 40), which is problematic because this trial did not always coincide with the true passing trial (by a maximum of two additional trials; see below).

For edge-population grackles, *Logan et al., 2023c*, used a sliding-window learning criterion, in which grackles were required to again make 17 out of the last 20 choices correctly, with the same minimum correct-choice counts for the previous two 10-trial sets, except that this criterion was assessed at every trial (from 20 onwards) rather than at the end of discrete sets. This second method is also problematic because a grackle can successfully reach criterion via a shift in the sliding-window *before* making a choice. For example, a grackle could make three wrong choices followed by 17 correct choices (i.e., 7/10 correct and 10/10 correct in the last two sets of 10 trials), and at the start of the next trial, the grackle will reach criterion because the summed choices now consist of 8/10 correct and at least 9/10 correct in the last two sets of 10 trials no matter their subsequent choice—see initial learning performance by bird 'Kel' for a real example (row 1816 in GitHub; as well as in *Logan et al., 2023c*). Moreover, the use of different learning criteria (fixed- and sliding-window) by Logan and colleagues in different populations represents a confound when populations are compared. Thus, our applied 17/20 learning criterion ensures our assessment of grackles' reinforcement learning is informative, straightforward, and consistent.

As a consequence of applying our 17/20 learning criterion, grackles can remain in initial and/or reversal learning beyond reaching criterion. These extra learning trials, however, already exist for some core- and middle-population grackles originally assessed via the fixed-window learning criterion ($N = 18$ grackles in initial learning [range: 1–2 extra trials]; $N = 13$ grackles in reversal learning [range: 1–2 extra trials]), as explained above. And our cleaning of the original data (see our Data_Processing.R script at GitHub) detected additional cases where grackles remained in-test despite meeting the applied criterion (fixed-window: $N = 1$ grackle in reversal learning for 11 extra trials; sliding-window: $N = 11$ grackles in initial learning [range: 1–10 extra trials]; $N = 7$ grackles in reversal learning [range: 1–14 extra trials]), presumably due to original experimenter oversight. Similarly, our data cleaning detected two grackles in the middle population that were passed by the original experimenters despite not meeting the assigned fixed-window learning criterion; instead, both chose 7/10 and 10/10 correct choices in the last two sets of 10 trials. Moreover, our data cleaning detected an additional four grackles in the core population that did not in fact meet the fixed-window learning

criterion because of incorrect trail numbers entered by the original experimenters e.g., skipping trial 24. In any case, in our study we verified: (i) our 17/20 learning criterion results in a similar proportion of male and female grackles experiencing extra initial learning trials (females, 15/17; males, 30/32); and (ii) our learning parameter estimations during initial learning remain relatively unchanged irrespective of whether we exclude or include extra initial learning trails (*Figure 2—figure supplement 1*). Thus, we are confident that any carryover effect of extra initial learning trials on grackles' reversal learning in our study is negligible if not nonexistent, and we therefore excluded extra learning trials.

## Statistical analyses

We analysed, processed, and visually present our data using, respectively, the 'rstan' (*Stan Development Team, 2020*), 'rethinking' (*McElreath, 2018*), and 'tidyverse' (*Wickham et al., 2019*) packages in R (*R Development Core Team, 2021*). We note our reproducible code is available at GitHub. We further note our reinforcement learning model, defined below, does not exclude cases—two males in the core, and one male in the middle population—where a grackle was dropped (due to time constraints) early on from reversal learning by the original experimenters: because individual-level $\varphi$ and $\lambda$ estimates can still be estimated irrespective of trial number; the certainty around the estimates will simply be wider (*McElreath, 2018*). Our Poisson models, however, do exclude these three cases for our modelling of reversal learning, to conserve estimation. Finally, we note that in Bayesian statistics, there is no strict lower limit of required sample size as the inferences do not rely on asymptotic assumptions. With inferences remaining valid in principle, low sample size will of course be reflected in rather uncertain posterior estimates. We further note all of our multi-level models use partial pooling on individuals (the random-effects structure), which is a regularisation technique that especially improves inference in case of low sample sizes (see Ch. 13 in *McElreath, 2018*). The full output from each of our models, which use weakly informative priors, is available in *Supplementary file 2a–d*, including posterior means and 89% highest posterior density intervals that convey the most likely values of our parameters over this range (*McElreath, 2018*).

## Poisson models

For our behavioural assay of reinforcement learning finishing trajectories, we used a multi-level Bayesian Poisson regression to quantify the effect(s) of sex and learning phase (initial versus reversal) on grackles' recorded number of trials to successfully finish each phase. This model was performed at both the population and across-population level, and accounted for individual differences among birds through the inclusion of individual-specific varying (i.e., random) effects.

For our behavioural assay of reinforcement learning choice-option switching, we used an identical Poisson model to that described above, to predict the total number of switches between the rewarded and unrewarded coloured tubes.

## Reinforcement learning model

We employed an adapted (from *Deffner et al., 2020*) multi-level Bayesian reinforcement learning model, to examine the influence of sex on grackles' initial and reversal learning. Our reinforcement learning model, defined below, allows us to link observed coloured tube choices to latent individual-level attraction updating, and to translate the influence of latent attractions (i.e., expected payoffs) into individual tube choice probabilities. As introduced above, we can reverse engineer which values of our two latent learning parameters—the information-updating rate $\varphi$ and the risk-sensitivity rate $\lambda$—most likely produce grackles' choice behaviour, by formulating our scientific model as a statistical model. Therefore, this computational method facilitates mechanistic insight into how multiple latent learning parameters simultaneously guide grackles' reinforcement learning (*McElreath, 2018*).

Our reinforcement learning model consists of two equations:

$$A_{i,j,t+1} = (1 - \phi_{k,l})A_{i,j,t} + \phi_{k,l}\pi_{i,j,t} \tag{1}$$

$$P(i)_{t+1} = \frac{\exp(\lambda_{k,l}A_{i,j,t})}{\sum\limits_{m=1}^{2} \exp(\lambda_{k,l}A_{m,j,t})} \tag{2}$$

*Equation 1* expresses how attraction $A$ to choice-option $i$ changes for an individual $j$ across time $(t + 1)$ based on their prior attraction to that choice-option ($A_{i,j,t}$) plus their recently experienced choice reward-payoffs ($\pi_{i,j,t}$), whilst accounting for the relative influence of recent reward-payoffs ($\phi_{k,l}$). As $\phi_{k,l}$ increases in value, so, too, does the rate of individual-level attraction updating based on reward-payoffs. Here, then, $\phi_{k,l}$ represents the information-updating rate. We highlight that the $k, l$ indexing (here and elsewhere) denotes that we estimate separate $\varphi$ parameters for each population ($k$=1 for core; $k = 2$ for middle; $k = 3$ for edge) and for each learning phase ($l = 1$ for females/initial, $l = 2$ for females/reversal; $l = 3$ for males/initial; $l = 4$ for males/reversal).

*Equation 2* is a *softmax* function that expresses the probability p that choice-option $i$ is selected in the next choice-round $(t + 1)$ as a function of the attractions $A$ and the parameter $\lambda_{k,l}$, which governs how much relative differences in attraction scores guide individual choice behaviour. In the reinforcement learning literature, the $\lambda$ parameter is known by several names—e.g., 'inverse temperature', 'exploration', or 'risk-appetite' (*Sutton and Barto, 2018*; *Chimento et al., 2022*)—since the higher its value the more deterministic the choice behaviour of an individual becomes (note $\lambda = 0$ generates random choice). In line with risk-sensitive foraging theory—which focuses on how animals evaluate and choose between distinct food options, and how such foraging decisions are influenced by pay-off variance i.e., risk associated with alternative foraging options (seminal reviews: *Bateson, 2002*; *Kacelnik and Bateson, 1996*)—we call $\lambda$ the risk-sensitivity rate, where higher values of $\lambda$ imply foragers are more sensitive to risk, seeking higher expected payoffs based on their prior experience, instead of randomly sampling alternative options.

From the above reinforcement learning model, then, we generate inferences about the effect of sex on $\phi_{k,l}$ and $\lambda_{k,l}$ from at least 1000 effective samples of the posterior distribution, at both the population- and across-population level. We note our reinforcement learning model also includes bird as a random effect (to account for repeated measures within individuals); however, for clarity, this parameter is omitted from our equations (but not our code: GitHub). Our reinforcement learning model does not, on the other hand, include trials where a grackle did not make a tube choice, as this measure cannot clearly speak to individual learning—e.g., satiation rather than any learning of 'appropriate' colour tube choice could be invoked as an explanation in such cases. Indeed, there are, admittedly, a number of intrinsic and extrinsic factors (e.g., temperament and temperature, respectively) that might bias grackles' tube choice behaviour, and, in turn, the output from our reinforcement learning model (*Webster and Rutz, 2020*). But the aim of such models is not to replicate the entire study system. Finally, we further note, while we exclude extra learning trials from all of our analyses (see above), our reinforcement learning model initiates estimation of $\phi$ and $\lambda$ during reversal learning, based on individual-level attractions encompassing all previous choices. This parameterisation ensures we precisely capture grackles' attraction scores up to the point of stimulus-reward reversal (for details, see our RL_Execution.R script at GitHub).

## Agent-based simulations: pre- and post-study

Prior to analysing our data, we used agent-based simulations to validate our reinforcement learning model (full details in our preregistration–see *Supplementary file 1*). In brief, the tube choice behaviour of simulants is governed by a set of rules identical to those defined by *Equations 1 and 2*, and we apply the exact same learning criterion for successfully finishing both learning phases. Crucially, this a priori model vetting verifies our reinforcement learning model can (i) detect simulated sex-effects and (ii) accurately recover simulated parameter values in both extreme and more realistic scenarios.

After model fitting, we used the same agent-based approach to forward simulate—i.e., simulate via the posterior distribution—synthetic learning trajectories by 'birds' via individual-level parameter estimates generated from our across-population reinforcement learning model. Specifically, maintaining the correlation structure among sex- and phase-specific learning parameters, we draw samples from the full or averaged random-effects multivariate normal distribution describing the inferred population of grackles. We use these post-study forward simulations to gain a better understanding of the implied consequences of the estimated sex differences in grackles' learning parameters (see *Figure 2* and associated main text; for an example of this approach in a different context, see *Deffner et al., 2020*). We note agent-based forward simulation does not assess goodness-of-fit—we assessed the fit of our model a priori in our preregistration (see *Supplementary file 1*)—but it does assess whether

one did a comprehensive job of uncovering the mechanistic basis of target behaviour(s), as detailed in the main text.

## Evolutionary model

To investigate the evolutionary significance of strategising risk-sensitive learning, we used algorithmic optimisation techniques (*Yu and Gen, 2010*; *Otto and Day, 2011*). Specifically, we construct an evolutionary model of grackle learning, to estimate how our learning parameters—the information-updating rate $\varphi$ and the risk-sensitivity rate $\lambda$—evolve in environments that systematically vary across two ecologically relevant (see main text) statistical properties: the rate of environmental stability $u$ and the rate of environmental stochasticity $s$. The environmental stability parameter $u$ represents the probability that behaviour leading to a food reward changes from one choice to the next. If $u$ is small, individuals encounter a world where they can expect the same behaviour to be adaptive for a relatively long time. As $u$ becomes larger, optimal behaviour can change multiple times within an individual's lifetime. The environmental stochasticity parameter $s$ describes the probability that, on any given day, optimal behaviour may not result in a food reward due to external causes specific to this day. If $s$ is small, optimal behaviour reliably produces rewards. As $s$ becomes larger, there is more and more daily 'noise' regarding which behaviour is rewarded.

We consider a population of fixed size with $N = 300$ individuals. Each generation, individual agents are born naïve and make $t = 1000$ binary foraging decisions resulting in a food reward (or not). Agents decide and learn about the world through reinforcement learning governed by their individual learning parameters, $\varphi$ and $\lambda$ (see *Equations 1 and 2*). Both learning parameters can vary continuously, corresponding to the infinite-alleles model from population genetics (*Otto and Day, 2011*). Over the course of their lifetime, agents collect food rewards, and the sum of rewards collected over the last 800 foraging decisions (or 'days') determines their individual fitness. We ignore the first 200 choices because selection should respond to the steady state of the environment, independently of initial conditions (*Otto and Day, 2011*).

To generate the next generation, we assume asexual, haploid reproduction, and use fitness-proportionate (or 'roulette wheel') selection to choose individuals for reproduction (*Yu and Gen, 2010*; *Otto and Day, 2011*). Here, juveniles inherit both learning parameters, $\varphi$ and $\lambda$, from their parent but with a small deviation (in random direction) due to mutation. Specifically, during each mutation event, a value drawn from zero-centered normal distributions $N(0, \mu_\phi)$ or $N(0, \mu_\lambda)$ is added to the parent value on the logit/log-scale to ensure parameters remain within allowed limits (between 0 and 1 for $\varphi$; positive for $\lambda$). The mutation parameters $\mu_\phi$ and $\mu_\lambda$ thus describe how much offspring values might deviate from parental values, which we set to 0.05. We restrict the risk-sensitivity rate $\lambda$ to the interval 0–15, because greater values result in identical choice behaviour. All results reported in the main text are averaged over the last 5000 generations of 10 independent 7000-generation simulations per parameter combination. This duration is sufficient to reach steady state in all cases.

## Acknowledgements

We thank Jean-François Gerard and Rachel Harrison for useful feedback on our study before data analyses; and James St Clair, Sue Healy, and Richard McElreath for similarly useful presubmission feedback. We also thank two anonymous reviewers for their thoughtful and useful feedback on our manuscript. We are further and most grateful to Richard McElreath for study and overall full support. And we thank all members, past and present, of the Grackle Project for collecting and making available, either via public archive or permissed advanced access, the data utilised herein (see Data provenance). We note this material uses illustrations from Vecteezy.com. Finally, we further note this material uses data from the eBird Status and Trends Project at the Cornell Lab of Ornithology, eBird.org. Any opinions, findings, and conclusions or recommendations expressed in this material are those of the authors and do not necessarily reflect the views of the Cornell Lab of Ornithology.

# Additional information

## Funding
 No other funding was received for this work.

## Author contributions
Alexis J Breen, Conceptualization, Data curation, Software, Formal analysis, Validation, Visualization, Methodology, Writing – original draft, Project administration, Writing – review and editing; Dominik Deffner, Conceptualization, Software, Formal analysis, Validation, Visualization, Methodology, Writing – review and editing

## Author ORCIDs
Alexis J Breen ⬤ https://orcid.org/0000-0002-2331-0920
Dominik Deffner ⬤ http://orcid.org/0000-0002-1649-3861

Reviewer #2 (Public Review): https://doi.org/10.7554/eLife.89315.3.sa1
Author Response https://doi.org/10.7554/eLife.89315.3.sa2

---

# Additional files

## Supplementary files
• Supplementary file 1. Study preregistration, including reinforcement learning model validation.

• Supplementary file 2. Supplementary tables. (a) Total-trials-in-test Poisson regression model output. (b) Total-choice-option-switches-in-test Poisson regression model output. (c) Bayesian reinforcement learning model information-updating rate $\varphi$ output. (d) Bayesian reinforcement learning model risk-sensitivity rate $\lambda$ output. For (a–d), both between- and across-population posterior means and corresponding 89% highest-posterior density intervals are reported for males, females, and male-female contrasts.

• MDAR checklist

## Data availability
All data (original and cleaned files) and code (for data munging, analyses, and figures) to reproduce our manuscript are available at our GitHub repository, (copy archived at *Breen, 2024*).

The following previously published datasets were used:

| Author(s) | Year | Dataset title | Dataset URL | Database and Identifier |
|---|---|---|---|---|
| Logan C | 2018 | Great-tailed grackle behavioral flexibility and problem solving experiments, Santa Barbara, CA USA 2014-2015 | https://knb.ecoinformatics.org/view/doi:10.5063/F13B5XBC | Knowledge Network for Biocomplexity, 10.5063/F13B5XBC |
| Logan C, McCune K, LeGrande-Rolls C, Marfori Z, Hubbard J, Lukas D | 2023 | Data: Implementing a rapid geographic range expansion - the role of behavior changes | https://knb.ecoinformatics.org/view/doi:10.5063/F1QZ28FH | Knowledge Network for Biocomplexity, 10.5063/F1QZ28FH |
| Logan C, Blaisdell A, Johnson-Ulrich Z, Lukas D, MacPherson M, Seitz B, Sevchik A, McCune K | 2021 | Data: Is behavioral flexibility manipulatable and, if so, does it improve flexibility and problem solving in a new context? | https://knb.ecoinformatics.org/view/doi:10.5063/F1862DWC | Knowledge Network for Biocomplexity, 10.5063/F1862DWC |

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
