## [Editor Report · eLife assessment]

This **important** study uses a multi-pronged empirical and theoretical approach to advance our understanding of animal cognition. It presents **convincing** data on how differences in learning relate to differences in the ways that male versus female animals cope with urban environments, and more generally how reversal learning may benefit animals in urban habitats.

---

## [Referee Report · Reviewer #2 (Public Review)]

Summary: The study is titled "Leading an urban invasion: risk-sensitive learning is a winning strategy", and consists of three different parts. First, the authors analyse data on initial and reversal learning in Grackles confronted with a foraging task, derived from three populations labeled as "core", "middle" and "edge" in relation to the invasion front. The suggested difference between study populations does not surface, but the authors do find support for a difference between male and female individuals. Secondly, the authors confirm that the proposed mechanism can actually generate patterns such as observed in the Grackle data through agent-based forward simulations. In the third part, the authors present an evolutionary model, in which they show that learning strategies, as observed in male Grackles, do evolve in simplified urban conditions including different levels of environmental stability and environmental stochasticity.

Strengths: The manuscript's strength is that it combines real learning data collected across different populations of the Great-tailed grackle (Quiscalus mexicanus) with theoretical approaches to better understand the processes with which grackles learn and how such learning processes might be advantageous during range expansion and invasion. Furthermore, the authors also take sex into account revealing that males, the dispersing sex, show better reversal learning through higher reward-payoff sensitivity. I also find it refreshing to see that the authors took the time to preregister their study to improve transparency especially regarding data analysis.

Weakness: The small sample size of grackles across populations increases uncertainty as to parameter estimates and the conclusions drawn from these estimates.

After revision, the introduction is appropriate, and in the methods, the authors take great care in explaining the rational behind decisions as to the selection of analysis methods and parameters. I very much appreciate that the authors took such care in revising their paper, the quality of which has now greatly improved.

---

## [Author Response]

The following is the authors’ response to the original reviews.

**eLife assessment**
This study uses a multi-pronged empirical and theoretical approach to advance our understanding of how differences in learning relate to differences in the ways that male versus female animals cope with urban environments, and more generally how reversal learning may benefit animals in urban habitats. The work makes an important contribution and parts of the data and analyses are solid, although several of the main claims are only partially supported or overstated and require additional support.
**Public Reviews:**

We thank the Editor and both Reviewers for their time and for their constructive evaluation of our manuscript. We worked to address each comment and suggestion offered by the Reviewers in our revision—please see our point-by-point responses below.

**Reviewer #1 (Public Review):**
Summary:In this highly ambitious paper, Breen and Deffner used a multi-pronged approach to generate novel insights on how differences between male and female birds in their learning strategies might relate to patterns of invasion and spread into new geographic and urban areas.The empirical results, drawn from data available in online archives, showed that while males and females are similar in their initial efficiency of learning a standard color-food association (e.g., color X = food; color Y = no food) scenario when the associations are switched (now, color Y = food, X = no food), males are more efficient than females at adjusting to the new situation (i.e., faster at 'reversal learning'). Clearly, if animals live in an unstable world, where associations between cues (e.g., color) and what is good versus bad might change unpredictably, it is important to be good at reversal learning. In these grackles, males tend to disperse into new areas before females. It is thus fascinating that males appear to be better than females at reversal learning. Importantly, to gain a better understanding of underlying learning mechanisms, the authors use a Bayesian learning model to assess the relative role of two mechanisms (each governed by a single parameter) that might contribute to differences in learning. They find that what they term 'risk sensitive' learning is the key to explaining the differences in reversal learning. Males tend to exhibit higher risk sensitivity which explains their faster reversal learning. The authors then tested the validity of their empirical results by running agent-based simulations where 10,000 computersimulated 'birds' were asked to make feeding choices using the learning parameters estimated from real birds. Perhaps not surprisingly, the computer birds exhibited learning patterns that were strikingly similar to the real birds. Finally, the authors ran evolutionary algorithms that simulate evolution by natural selection where the key traits that can evolve are the two learning parameters. They find that under conditions that might be common in urban environments, high-risk sensitivity is indeed favored.Strengths:The paper addresses a critically important issue in the modern world. Clearly, some organisms (some species, some individuals) are adjusting well and thriving in the modern, human-altered world, while others are doing poorly. Understanding how organisms cope with human-induced environmental change, and why some are particularly good at adjusting to change is thus an important question.The comparison of male versus female reversal learning across three populations that differ in years since they were first invaded by grackles is one of few, perhaps the first in any species, to address this important issue experimentally.Using a combination of experimental results, statistical simulations, and evolutionary modeling is a powerful method for elucidating novel insights.

Thank you—we are delighted to receive this positive feedback, especially regarding the inferential power of our analytical approach.

Weaknesses:The match between the broader conceptual background involving range expansion, urbanization, and sex-biased dispersal and learning, and the actual comparison of three urban populations along a range expansion gradient was somewhat confusing. The fact that three populations were compared along a range expansion gradient implies an expectation that they might differ because they are at very different points in a range expansion. Indeed, the predicted differences between males and females are largely couched in terms of population differences based on their 'location' along the rangeexpansion gradient. However, the fact that they are all urban areas suggests that one might not expect the populations to differ. In addition, the evolutionary model suggests that all animals, male or female, living in urban environments (that the authors suggest are stable but unpredictable) should exhibit high-risk sensitivity. Given that all grackles, male and female, in all populations, are both living in urban environments and likely come from an urban background, should males and females differ in their learning behavior? Clarification would be useful.

Thank you for highlighting a gap in clarity in our conceptual framework. To answer the Reviewer’s question—yes, even with this shared urban ‘history’, it seems plausible that males and females could differ in their learning. For example, irrespective of population membership, such sex differences could come about via differential reliance on learning strategies mediated by an interaction between grackles’ polygynous mating system and malebiased dispersal system, as we discuss in L254–265 (now L295–306). Population membership might, in turn, differentially moderate the magnitude of any such sex-effect since an edge population, even though urban, could still pose novel challenges—for example, by requiring grackles to learn novel daily temporal foraging patterns such as when and where garbage is collected (grackles appear to track this food resource: Rodrigo et al. 2021 [DOI: 10.1101/2021.06.14.448443]). We now introduce this important conceptual information— please see L89–96.

Reinforcement learning mechanisms:Although the authors' title, abstract, and conclusions emphasize the importance of variation in 'risk sensitivity', most readers in this field will very possibly misunderstand what this means biologically. Both the authors' use of the term 'risk sensitivity' and their statistical methods for measuring this concept have potential problems.

Please see our below responses concerning our risk-sensitivity term.

First, most behavioral ecologists think of risk as predation risk which is not considered in this paper. Secondarily, some might think of risk as uncertainty. Here, as discussed in more detail below, the 'risk sensitivity' parameter basically influences how strongly an option's attractiveness affects the animal's choice of that option. They say that this is in line with foraging theory (Stephens and Krebs 2019) where sensitivity means seeking higher expected payoffs based on prior experience. To me, this sounds like 'reward sensitivity', but not what most think of as 'risk sensitivity'. This problem can be easily fixed by changing the name of the term.

We apologise for not clearly introducing the field of risk-sensitive foraging, which focuses on how animals evaluate and choose between distinct food options, and how such foraging decisions are influenced by pay-off variance i.e., risk associated with alternative foraging options (seminal reviews: Bateson 2002 [DOI: 10.1079/PNS2002181]; Kacelnik & Bateson 1996 [DOI: 10.1093/ICB/36.4.402]). We have added this information to our manuscript in L494–497. We further apologise for not clearly explaining how our lambda parameter estimates such risk-sensitive foraging. To do so here, we need to consider our Bayesian reinforcement learning model in full. This model uses observed choice-behaviour during reinforcement learning to infer our phi (information-updating) and lambda (risksensitivity) learning parameters. Thus, payoffs incurred through choice simultaneously influence estimation of each learning parameter—that is, in a sense, they are *both* sensitive to rewards. But phi and lambda differentially direct any reward sensitivity back on choicebehaviour due to their distinct definitions. Glossing over the mathematics, for phi, stronger reward sensitivity (bigger phi values) means faster internal updating about stimulus-reward pairings, which translates behaviourally into faster learning about ‘what to choose’. For lambda, stronger reward sensitivity (bigger lambda values) means stronger internal determinism about seeking the non-risk foraging option (i.e., the one with the higher expected payoffs based on prior experience), which translates behaviourally into less choice-option switching i.e., ‘playing it safe’. We hope this information, which we have incorporated into our revised manuscript (please see L153–161), clarifies the rationale and mechanics of our reinforcement learning model, and why lamba measures risk-sensitivity.

In addition, however, the parameter does not measure sensitivity to rewards per se - rewards are not in equation 2. As noted above, instead, equation 2 addresses the sensitivity of choice to the attraction score which can be sensitive to rewards, though in complex ways depending on the updating parameter. Second, equations 1 and 2 involve one specific assumption about how sensitivity to rewards vs. to attraction influences the probability of choosing an option. In essence, the authors split the translation from rewards to behavioral choices into 2 steps. Step 1 is how strongly rewards influence an option's attractiveness and step 2 is how strongly attractiveness influences the actual choice to use that option. The equation for step 1 is linear whereas the equation for step 2 has an exponential component. Whether a relationship is linear or exponential can clearly have a major effect on how parameter values influence outcomes. Is there a justification for the form of these equations? The analyses suggest that the exponential component provides a better explanation than the linear component for the difference between males and females in the sequence of choices made by birds, but translating that to the concepts of information updating versus reward sensitivity is unclear. As noted above, the authors' equation for reward sensitivity does not actually include rewards explicitly, but instead only responds to rewards if the rewards influence attraction scores. The more strongly recent rewards drive an update of attraction scores, the more strongly they also influence food choices. While this is intuitively reasonable, I am skeptical about the authors' biological/cognitive conclusions that are couched in terms of words (updating rate and risk sensitivity) that readers will likely interpret as concepts that, in my view, do not actually concur with what the models and analyses address.

To answer the Reviewer’s question—yes, these equations are very much standard and the canonical way of analysing individual reinforcement learning (see: Ch. 15.2 in Computational Modeling of Cognition and Behavior by Farrell & Lewandowsky 2018 [DOI: 10.1017/CBO9781316272503]; McElreath et al. 2008 [DOI: 10.1098/rstb/2008/0131]; Reinforcement Learning by Sutton & Barto 2018). To provide a “justification for the form of these equations'', equation 1 describes a convex combination of previous values and recent payoffs. Latent values are updated as a linear combination of both factors, there is no simple linear mapping between payoffs and behaviour as suggested by the reviewer. Equation 2 describes the standard softmax link function. It converts a vector of real numbers (here latent values) into a simplex vector (i.e., a vector summing to 1) which represents the probabilities of different outcomes. Similar to the logit link in logistic regression, the softmax simply maps the model space of latent values onto the outcome space of choice probabilities which enter the categorial likelihood distribution. We can appreciate how we did not make this clear in our manuscript by not highlighting the standard nature of our analytical approach—we now do so in our revised manuscript (please see L148–149). As far as what our reinforcement learning model measures, and how it relates cognition and behaviour, please see our previous response.

To emphasize, while the authors imply that their analyses separate the updating rate from 'risk sensitivity', both the 'updating parameter' and the 'risk sensitivity' parameter influence both the strength of updating and the sensitivity to reward payoffs in the sense of altering the tendency to prefer an option based on recent experience with payoffs. As noted in the previous paragraph, the main difference between the two parameters is whether they relate to behaviour linearly versus with an exponential component.

Please see our two earlier responses on the mechanics of our reinforcement learning model.

Overall, while the statistical analyses based on equations (1) and (2) seem to have identified something interesting about two steps underlying learning patterns, to maximize the valuable conceptual impact that these analyses have for the field, more thinking is required to better understand the biological meaning of how these two parameters relate to observed behaviours, and the 'risk sensitivity' parameter needs to be re-named.

Please see our earlier response to these suggestions.

Agent-based simulations:The authors estimated two learning parameters based on the behaviour of real birds, and then ran simulations to see whether computer 'birds' that base their choices on those learning parameters return behaviours that, on average, mirror the behaviour of the real birds. This exercise is clearly circular. In old-style, statistical terms, I suppose this means that the R-square of the statistical model is good. A more insightful use of the simulations would be to identify situations where the simulation does not do as well in mirroring behaviour that it is designed to mirror.

Based on the Reviewer’s summary of agent-based forward simulation, we can see we did a poor job explaining the inferential value of this method—we apologise. Agent-based forward simulations are posterior predictions, and they provide insight into the implied model dynamics and overall usefulness of our reinforcement learning model. R-squared calculations are retrodictive, and they say nothing about the causal dynamics of a model. Specifically, agent-based forward simulation allows us to ask—what would a ‘new’ grackle ‘do’, given our reinforcement learning model parameter estimates? It is important to ask this question because, in parameterising our model, we may have overlooked a critical contributing mechanism to grackles’ reinforcement learning. Such an omission is invisible in the raw parameter estimates; it is only betrayed by the parameters *in actu*. Agent-based forward simulation is ‘designed’ to facilitate this call to action—not to mirror behavioural results. The simulation has no *apriori* ‘opinion’ about computer ‘birds’ behavioural outcomes; rather, it simply assigns these agents random phi and lambda draws (whilst maintaining their correlation structure), and tracks their reinforcement learning. The exercise only appears circular *if* no critical contributing mechanism(s) went overlooked—in this case computer ‘birds’ should behave similar to real birds. A disparate mapping between computer ‘birds’ and real birds, however, would mean more work is needed with respect to model parameterisation that captures the causal, mechanistic dynamics behind real birds’ reinforcement learning (for an example of this happening in the human reinforcement learning literature, see Deffner et al. 2020 [DOI: 10.1098/rsos.200734]). In sum, agent-based forward simulation does not access goodness-of-fit—we assessed the fit of our model *apriori* in our preregistration (https://osf.io/v3wxb)—but it does assess whether one did a comprehensive job of uncovering the mechanistic basis of target behaviour(s). We have worked to make the above points on the method and the insight afforded by agent-based forward simulation explicitly clear in our revision—please see L192–207 and L534–537.

**Reviewer #2 (Public Review):**
Summary:The study is titled "Leading an urban invasion: risk-sensitive learning is a winning strategy", and consists of three different parts. First, the authors analyse data on initial and reversal learning in Grackles confronted with a foraging task, derived from three populations labeled as "core", "middle" and "edge" in relation to the invasion front. The suggested difference between study populations does not surface, but the authors do find moderate support for a difference between male and female individuals. Secondly, the authors confirm that the proposed mechanism can actually generate patterns such as those observed in the Grackle data. In the third part, the authors present an evolutionary model, in which they show that learning strategies as observed in male Grackles do evolve in what they regard as conditions present in urban environments.Strengths:The manuscript's strength is that it combines real learning data collected across different populations of the Great-tailed grackle (Quiscalus mexicanus) with theoretical approaches to better understand the processes with which grackles learn and how such learning processes might be advantageous during range expansion. Furthermore, the authors also take sex into account revealing that males, the dispersing sex, show moderately better reversal learning through higher reward-payoff sensitivity. I also find it refreshing to see that the authors took the time to preregister their study to improve transparency, especially regarding data analysis.

Thank you—we are pleased to receive this positive evaluation, particularly concerning our efforts to improve scientific transparency via our study’s preregistration (https://osf.io/v3wxb).

Weaknesses:One major weakness of this manuscript is the fact that the authors are working with quite low sample sizes when we look at the different populations of edge (11 males & 8 females), middle (4 males & 4 females), and core (17 males & 5 females) expansion range. Although I think that when all populations are pooled together, the sample size is sufficient to answer the questions regarding sex differences in learning performance and which learning processes might be used by grackles but insufficient when taking the different populations into account.

In Bayesian statistics, there is no strict lower limit of required sample size as the inferences do not rely on asymptotic assumptions. With inferences remaining valid in principle, low sample size will of course be reflected in rather uncertain posterior estimates. We note all of our multilevel models use partial pooling on individuals (the random-effects structure), which is a regularisation technique that generally reduces the inference constraint imposed by a low sample size (see Ch. 13 in Statistical Rethinking by Richard McElreath [PDF: https://bit.ly/3RXCy8c]). We further note that, in our study preregistration (https://osf.io/v3wxb), we formally tested our reinforcement learning model for different effect sizes of sex on learning for both target parameters (phi and lambda) across populations, using a similarly modest N (edge: 10 M, 5 F; middle: 22 M, 5 F ; core: 3 M, 4 F) to our actual final N, that we anticipated to be our final N at that time. This apriori analysis shows our reinforcement learning model: (i) detects sex differences in phi values >= 0.03 and lambda values >= 1; and (ii) infers a null effect for phi values < 0.03 and lambda values < 1 i.e., very weak simulated sex differences (see Figure 4 in https://osf.io/v3wxb). Thus, both of these points together highlight how our reinforcement learning model allows us to say that across-population null results are not just due to small sample size. Nevertheless the Reviewer is not wrong to wonder whether a bigger N might change our population-level results (it might; so might muchneeded population replicates—see L310), but our Bayesian models still allow us to learn a lot from our current data. We now explain this in our revised manuscript—please see L452–457.

Another weakness of this manuscript is that it does not set up the background well in the introduction. Firstly, are grackles urban dwellers in their natural range and expand by colonising urban habitats because they are adapted to it? The introduction also fails to mention why urban habitats are special and why we expect them to be more challenging for animals to inhabit. If we consider that one of their main questions is related to how learning processes might help individuals deal with a challenging urban habitat, then this should be properly introduced.

In L74–75 (previously L53–56) we introduce that the estimated historical niche of grackles is urban environments, and that shifts in habitat breadth—e.g., moving into more arid, agricultural environments—is the estimated driver of their rapid North American colonisation.We hope this included information sufficiently answers the Reviewer’s question. We have worked towards flushing out how urban-imposed challenges faced by grackles, such as the wildlife management efforts introduced in L64–65 (now L85–86), may apply to animals inhabiting urban environments more broadly; for example, we now include an entire paragraph in our Introduction detailing how urban environments may be characterised differently to nonurban environments, and thus why they are perhaps more challenging for animals to inhabit— please see L56–71.

Also, the authors provide a single example of how learning can differ between populations from more urban and more natural habitats. The authors also label the urban dwellers as the invaders, which might be the case for grackles but is not necessarily true for other species, such as the Indian rock agama in the example which are native to the area of study. Also, the authors need to be aware that only male lizards were tested in this study. I suggest being a bit more clear about what has been found across different studies looking at: (1) differences across individuals from invasive and native populations of invasive species and (2) differences across individuals from natural and urban populations.

We apologise for not including more examples of such learning differences. We now include three examples (please see L43–49), and we are careful to call attention to the fact that these data cover both resident urban and non-urban species as well as urban invasive species (please see L49–50). We also revised our labelling of the lizard species (please see L44). We are aware only male lizards were tested but this information is not relevant to substantiating our use of this study; that is, to highlight that learning can differ between urbandwelling and non-urban counterparts. We hope the changes we did make to our manuscript satisfy the Reviewer’s general suggestion to add biological clarity.

Finally, the introduction is very much written with regard to the interaction between learning and dispersal, i.e. the 'invasion front' theme. The authors lay out four predictions, the most important of which is No. 4: "Such sex-mediated differences in learning to be more pronounced in grackles living at the edge, rather than the intermediate and/or core region of their range." The authors, however, never return to this prediction, at least not in a transparent way that clearly pronounces this pattern not being found. The model looking at the evolution of risk-sensitive learning in urban environments is based on the assumption that urban and natural environments "differ along two key ecological axes: environmental stability *u* (How often does optimal behaviour change?) and environmental stochasticity *s* (How often does optimal behaviour fail to pay off?). Urban environments are generally characterised as both stable (lower *u*) and stochastic (higher *s*)". Even though it is generally assumed that urban environments differ from natural environments the authors' assumption is just one way of looking at the differences which have generally not been confirmed and are highly debated. Additionally, it is not clear how this result relates to the rest of the paper: The three populations are distinguished according to their relation to the invasion front, not with respect to a gradient of urbanization, and further do not show a meaningful difference in learning behaviour possibly due to low sample sizes as mentioned above.

Thank you for highlighting a gap in our reporting clarity. We now take care to transparently report our null result regarding our fourth prediction; more specifically, that we did not detect credible population-level differences in grackles’ learning (please see L130). Regarding our evolutionary model, we agree with the Reviewer that this analysis is only one way of looking at the interaction between learning phenotype and apparent urban environmental characteristics. Indeed, in L282–288 (now L325–329) we state: “Admittedly, our evolutionary model is not a complete representation of urban ecology dynamics. Relevant factors—e.g., spatial dynamics and realistic life histories—are missed out. These omissions are tactical ones. Our evolutionary model solely focuses on the response of reinforcement learning parameters to two core urban-like (or not) environmental statistics, providing a baseline for future study to build on”. But we can see now that ‘core’ is too strong a word, and instead ‘supposed’, ‘purported’ or ‘theorised’ would be more accurate—we have revised our wording throughout our manuscript to say as much (please see, for example, L24; L56; L328). We also further highlight the preliminary nature of our evolutionary model, in terms of allowing a narrow but useful first-look at urban eco-evolutionary dynamics—please see L228–232. Finally, we now detail the theorised characteristics of urban environments in our Introduction (rather than in our Results; please see L56–71), and we hope that by doing so, how our evolutionary results relate to the rest of our paper is now better set up and clear.

In conclusion, the manuscript was well written and for the most part easy to follow. The format of eLife having the results before the methods makes it a bit harder to follow because the reader is not fully aware of the methods at the time the results are presented. It would, therefore, be important to more clearly delineate the different parts and purposes. Is this article about the interaction between urban invasion, dispersal, and learning? Or about the correct identification of learning mechanisms? Or about how learning mechanisms evolve in urban and natural environments? Maybe this article can harbor all three, but the borders need to be clear. The authors need to be transparent about what has and especially what has not been found, and be careful to not overstate their case.

Thank you, we are pleased to read that the Reviewer found our manuscript to be generally digestible. We have worked to add further clarity, and to tempter our tone (please see our above and below responses).

**Reviewer #1 (Recommendations For The Authors):**
Several of the results are based on CIs that overlap zero. Tone these down somewhat.

We apologise for overstating our results, which we have worked to tone down in our revision. For instance, in L185–186 we now differentiate between estimates that did or did not overlap zero (please also see our response to Reviewer 2 on this tonal change). We note we do not report confidence intervals (i.e., the range of values expected to contain the true estimate if one redoes the study/analysis many times). Rather, we report 89% highest posterior density intervals (i.e., the most likely values of our parameters over this range). We have added this definition in L459, to improve clarity.

The literature review suggesting that urban environments are more unpredictable is not convincing. Yes, they have more noise and light pollution and more cars and planes, but does this actually relate to the unpredictability of getting a food reward when you choose an option that usually yields rewards?

To answer the Reviewer’s question—yes. But we can see that by not including empirical examples from the literature, we did a poor job of arguing such links. In L43–49 we now give three empirical examples; more specifically, we state: “[...] experimental data show the more variable are traffic noise and pedestrian presence, the more negative are such human-driven effects on birds' sleep (Grunst et al., 2021), mating (Blickley et al., 2012), and foraging behaviour (Fernández-Juricic, 2000).” We note we now detail such apparently stable but stochastic urban environmental characteristics in our Introduction rather than our Results section, to hopefully improve the clarity of our manuscript (please see L56–71). We further note that we cite three literature reviews—not one—suggesting urban environments are stable in certain characteristics and more unpredictable in others (please see L59–60). Finally, we appreciate such characterisation is not certain, and so in our revision we have qualified all writing about this potential dynamic with words such as “apparent”, “supposed”, “theorised”,“hypothesised” etc.

It would be interesting to see if other individual traits besides sex affect their learning/reversal learning ability and/or their learning parameters. Do you have data on age, size, condition, or personality? Or, the habitat where they were captured?

We do not have these data. But we agree with the Reviewer that examining the potential influence of such covariates on grackles’ reinforcement learning would be interesting in future study, especially habitat characteristics (please see L306–309).

For most levels of environmental noise, there appears to be an intermediate maximum for the relationship between environmental stability and the risk sensitivity parameter. What does this mean?

There is indeed an intermediate maximum for certain values of environmental stochasticity (although the differences are rather small). The most plausible reason for this is that for very stable environments, simulated birds essentially always “know” the rewarded solution and never need to “relearn” behaviour. In this case, differences in latent values will tend to be large (because they consistently get rewarded for the same option), and different lambda values (in the upper range) will produce the same choice behaviour, which results in very weak selection. While in very unstable environments, optimal choice behaviour should be more exploratory, allowing learners to track frequently-changing environments. We now note this pattern in L240–248.

**Reviewer #2 (Recommendations For The Authors):**
L2: I'd encourage the authors to reconsider the term "risk-sensitive learning", at least in the title. It's not apparent to me how 'risk' relates to the investigated foraging behaviour. Elsewhere, risk-reward sensitivity is used which may be a better term.

We apologise for not clearly introducing the field of risk-sensitive foraging, which focuses on how animals evaluate and choose between distinct food options, and how such foraging decisions are influenced by pay-off variance i.e., risk associated with alternative foraging options (seminal reviews: Bateson 2002 [DOI: 10.1079/PNS2002181]; Kacelnik & Bateson 1996 [DOI: 10.1093/ICB/36.4.402]). We have added this information to our manuscript in L494–497. In explaining our reinforcement model, we also now detail how risk relates to foraging behaviour. Specifically, in L153–161 we now state: “Both learning parameters capture individual-level internal response to incurred reward-payoffs, but they differentially direct any reward sensitivity back on choice-behaviour due to their distinct definitions (full mathematical details in Materials and methods). For *Φ*, stronger reward sensitivity (bigger values) means faster internal updating about stimulus-reward pairings, which translates behaviourally into faster learning about ‘what to choose’. For *λ*, stronger reward sensitivity (bigger values) means stronger internal determinism about seeking the nonrisk foraging option (i.e., the one with the higher expected payoffs based on prior experience), which translates behaviourally into less choice-option switching i.e., ‘playing it safe’.” We hope this information clarifies why lamba measures risk-sensitivity, and why we continue to use this term.

L1-3: The title is a bit misleading with regard to the empirical data. From the data, all that can be said is that male grackles relearn faster than females. Any difference between populations actually runs the other way, with the core population exhibiting a larger difference between males and females than the mid and edge populations.

It is customary for a manuscript title to describe the full scope of the study. In our study, we have empirical data, cognitive modelling, and evolutionary simulations of the background theory all together. And together these analytical approaches show: (1) across three populations, male grackles—the dispersing sex in this historically urban-dwelling and currently urban-invading species—outperform female counterparts in reversal learning; (2) they do this via risk-sensitive learning, so they’re more sensitive to relative differences in reward payoffs and choose to stick with the ‘safe’ i.e., rewarding option, rather than continuing to ‘gamble’ on an alternative option; and (3) risk-sensitive learning should be favoured in statistical environments characterised by purported urban dynamics. So, we do not feel our title “Leading an urban invasion: risk-sensitive learning is a winning strategy” is misleading with regard to our empirical data; it just doesn’t summarise only our empirical data. Finally, as we now state in L312–313, we caution against speculating about any between-population variation, as we did not infer any meaningful behavioural or mechanistic population-level differences.

L13: "Assayed", is that correctly put, given that the authors did not collect the data?

Merrian-Webster defines assay as “to analyse” or “examination or determination as to characteristics”, and so to answer the Reviewer’s question—yes, we feel this is correctly put. We note we explicitly introduce in L102–103 that we did not collect the data, and we have an explicit “Data provenance” section in our methods (please see L342–347).

L42-46: The authors provide a single example of how learning can differ between populations from more urban and more natural habitats. I would like to point out that many of these studies do not directly confirm that the ability in question has indeed led to the success of the species tested (e.g. show fitness consequences). Then the authors could combine these insights to form a solid prediction for the grackles. As of now, this looks like cherry-picking supportive literature without considering negative results.Here are some references that might be helpful in identifying relevant literature to cite:Szabo, B., Damas-Moreira, I., & Whiting, M. J. (2020). Can cognitive ability give invasive species the means to succeed? A review of the evidence. Frontiers in Ecology and Evolution, 8, 187.Griffin AS, Tebbich S, Bugnyar T, 2017. Animal cognition in a human-dominated world. Anim Cogn 20(1):1-6.Kark, S., Iwaniuk, A., Schalimtzek, A., & Banker, E. (2007). Living in the city: Can anyone become an "urban exploiter"? Journal of Biogeography, 34(4), 638-651.

We apologise for not including more examples of such learning differences. We now include three examples (please see L43–49). We are aware that direct evidence of fitness consequences is entirely lacking in the scientific literature on cognition and successful urban invasion; hence why such data is not present in our paper. But we now explicitly point out a role for likely fitness-affecting anthropogenic disturbances on sleep, mate, and foraging behaviour on animals inhabiting urban environments (please see L63–68). We hope these new data bolster our predictions for our grackles. Finally, the Reviewer paints a (in our view) inaccurate picture of our use of available literature. Nevertheless, to address their comment, we now highlight a recent meta-analysis advocating for further research to confirm apparent‘positive’ trends between animal ‘smarts’ and successful ‘city living’ (please see L43).

L64: Is their niche historically urban, or have they recently moved into urban areas?

In L74–75 (previously L53–56) we introduce that the estimated historical niche of grackles is urban environments, and that shifts in habitat breadth—e.g., moving into more arid, agricultural environments—is the estimated driver of their rapid North American colonisation. We hope this included information sufficiently answers the Reviewer’s question.

L66-67: This is an important point that is however altogether missing from the discussion.

We thank the Reviewer for highlighting a gap in our discussion regarding populationlevel differences in grackles’ reinforcement learning. In L310–312 we now state: “The lack of spatial replicates in the existing data set used herein inherently poses limitations on inference. Nevertheless, the currently available data do not show meaningful population-level behavioural or mechanistic differences in grackles’ reinforcement learning, and we should thus be cautious about speculating on between-population variation”.

L68-71: The paper focuses on cognitive ability. The whole paragraph sets up the prediction of why male grackles should be better learners due to their dispersal behaviour. This example, however, focuses on aggression, not cognition. Here is a study showing differences in learning in male and female mynas that might be better suited:Federspiel IG, Garland A, Guez D, Bugnyar T, Healy SD, Güntürkün O, Griffin AS, 2017. Adjusting foraging strategies: a comparison of rural and urban common mynas (Acridotheres tristis). Anim Cogn 20(1):65-74.

We thank the Reviewer for suggesting this paper. We feel it is better suited to substantiating our point in the Discussion about reversal learning not being indicative of cognitive ability—please see L276–277.

L73: Generally, I suggest not writing "for the first time" as this is not a valid argument for why a study should be conducted. Furthermore, except for replication studies, most studies investigate questions that are novel and have not been investigated before.

The Reviewer makes a fair point—we have removed this statement.

L80-81: Here again, this is left undiscussed later on.

By ‘this’ we assume the Reviewer is referring to our hypothesis, which is that sex differences in dispersal are related to sex differences in learning in an urban invader— grackles. At the beginning of our Discussion, we state how we found support for this hypothesis (please see L250–261); and in our ‘Ideas and speculation’ section, we discuss how these hypothesis-supporting data fit into the literature more broadly (please see L294–331). We feel this is therefore sufficiently discussed.

L77-81: This sentence is very long and therefore hard to read. I suggest trying to split it into at least 2 separate sentences which would improve readability.

Per the Reviewer’s useful suggestion, we have split this sentence into two separate sentences—please see L97–115.

L83: Please explain choice-option switches. I am not aware of what that is and it should be explained at first mention.

We apologise for this operational oversight. We now include a working definition of speed *and* choice-option switches at first mention. Specifically, in L107–108 we state: “[...] we expect male and female grackles to differ across at least two reinforcement learning behaviours: speed (trials to criterion) and choice-option switches (times alternating between available stimuli)”.

L83-87: Again, a very long sentence. Please split.

We thank the Reviewer for their suggestion. In this case we feel it is important to not change our sentence structure because we want our prediction statements to match between our manuscript and our preregistration.

L96-97: Important to not overstate this. It merely demonstrates the potential of the proposed (not detected) mechanism to generate the observed data.

As in any empirical analysis, our drawn conclusions depend on causal assumptions about the mechanisms generating behaviour (Pearl, J. (2009). Causality). Therefore, we “detected” specific learning mechanisms assuming a certain generative model, namely reinforcement learning. As there is overwhelming evidence for the widespread importance of value-based decision making and Rescorla-Wagner updating rules across numerous different animals (Sutton & Barto (2018) Reinforcement Learning), we would argue that this assumed model is highly plausible in our case. Still, we changed the text to “inferred” instead of“detected” learning mechanisms to account for this concern—please see L123–124.

L99: "urban-like settings" again a bit confusing. The authors talk about invasion fronts, but now also about an urbanisation gradient. Is the main difference between the size and the date of establishment, or is there additionally a gradient in urbanisation to be considered?

We now include a paragraph in our Introduction detailing apparent urban environmental characteristics (please see 56–71), and we now refer to this dynamic specifically when we define urban-like settings (please see L126–127). To answer the Reviewer’s question—we consider both differences. Specifically, we consider the time since population establishment in our paper (with respect to our behavioural and mechanistic modelling), as well as how statistical environments that vary in how similar they are to apparently characteristically urban-like environments, might favour particular learning phenotypes (with respect to our evolutionary modelling). We hope the edits to our Introduction as a whole now make both of the aims clear.

L11-112: Above the authors talk about a comparable number of switches (10.5/15=0.7), and here of fewer number of switches (25/35=0.71), even though the magnitude of the difference is almost identical and actually runs the other way. The authors are probably misled by their conservative priors, which makes the difference appear greater in the second case than in the first. Using flat priors would avoid this particular issue.

Mathematically, the number of trials-to-finish and the number of choice-optionswitches are both a Poisson distributed outcome with rate λ (we note lambda here is not our risk-sensitivity parameter; just standard notation). As such, our Poisson models infer the rate of these outcomes by sex and phase—not the ratio of these outcomes by sex and phase. So comparing the magnitude of divided medians of choice-option-switches between the sexes by phase is not a meaningful metric with respect to the distribution of our data, as the Reviewer does above. For perspective, 1 vs. 2 switches provides much less information about the difference in rates of a Poisson distribution than 50 vs 100 (for the former, no difference would be inferred; for the latter, it would), but both exhibit a 1:2 ratio. To hopefully prevent any such further confusion, and to focus on the fact that our Poisson models estimate the expected value i.e., the mean, we now report and graph (please see Fig. 2) mean and not median trialsto-finish and total-switch-counts. Finally, we can see that our use of the word “conservative” to describe our weakly informative priors is confusing, because conservative could mean either strong priors with respect to expected effect size (not our parameterisation) or weak priors with respect to such assumptions (our parameterisation). To address this lack of clarity, we now state that we use “weakly informative priors” in L457–458.

L126: It is not clear what risk sensitivity means in the context of these experiments.

Thank you for pointing out our lack of clarity. In L153–161 we now state: “Both learning parameters capture individual-level internal response to incurred reward-payoffs, but they differentially direct any reward sensitivity back on choice-behaviour due to their distinct definitions (full mathematical details in Materials and methods). For *Φ*, stronger reward sensitivity (bigger values) means faster internal updating about stimulus-reward pairings, which translates behaviourally into faster learning about ‘what to choose’. For *λ*, stronger reward sensitivity (bigger values) means stronger internal determinism about seeking the nonrisk foraging option (i.e., the one with the higher expected payoffs based on prior experience), which translates behaviourally into less choice-option switching i.e., ‘playing it safe’.” We hope this information clarifies what risk sensitivity means and measures, with respect to our behavioural experiments.

L128-129: I find this statement too strong. A plethora of other mechanisms could produce similar patterns, and you cannot exclude these by way of your method. All you can show is whether the mechanism is capable of producing broadly similar outcomes as observed

In describing the inferential value of our reinforcement learning model, we now qualify that the insight provided is of course conditional on the model, which is tonally accurate. Please see L161.

L144: As I have already mentioned above, here is the first time we hear about unpredictability related to urban environments. I suggest clearly explaining in the introduction how urban and natural environments are assumed to be different which leads to animals needing different cognitive abilities to survive in them which should explain why some species thrive and some species die out in urbanised habitats.

Thank you for this suggestion. We now include a paragraph in our Introduction detailing as much—please see L56–71.

L162: "almost entirely above zero" again, this is worded too strongly.

In reporting our lambda across-population 89% HPDI contrasts in L185–186, we now state: “[...] across-population contrasts that lie mostly above zero in initial learning, and entirely above zero in reversal learning”. Our previous wording stated: ““[...] across-population contrasts that lie almost entirely above zero”. The Reviewer was correct to point out that this previous wording was too strong if we considered the contrasts together, as, indeed, we find the range of the contrast in initial learning does minimally overlap zero (L: -0.77; U: 5.61), while the range of the contrast in reversal learning does not (L: 0.14; U: 4.26). This rephrasing is thus tonally accurate.

L178-179: I think it should be said instead that the model accounts well for the observed data.

We have rephrased in line with the Reviewer’s suggestion, now stating in L217–218 that “Such quantitative replication confirms our reinforcement learning model results sufficiently explain our behavioural sex-difference data.”

L188-190: I am not convinced this is a general pattern. It is quite a bold claim that I don't find to be supported by the citations. Why should biotic and abiotic factors differ in how they affect behavioural outcomes? Also, events in urban environments such as weekend/weekday could lead to highly regular optimal behaviour changes.

Please see our response to Reviewer 1 on this point. We note we now touch on such regular events in L94–96.

L209-211: The first sentence is misleading. The authors have found that males and females differ in 'risk sensitivity', that their learning model can fit the data rather well, and that under certain, not necessarily realistic assumptions, the male learning type is favoured by natural selection in urban environments. A difference between core, middle, and edge habitats however is barely found, and in fact seems to run the other way than expected.

In our study, we found: (1) across three populations, male grackles—the dispersing sex in this historically urban-dwelling and currently urban-invading species—outperform female counterparts in reversal learning; (2) they do this via risk-sensitive learning, so they’re more sensitive to relative differences in reward payoffs and choose to stick with the ‘safe’ i.e., rewarding option, rather than continuing to ‘gamble’ on an alternative option; (3) we are sufficiently certain risk-sensitive learning generates our sex-difference data, as our agentbased forward simulations replicate our behavioural results (not because our model ‘fits’ the data, but because we inferred meaningful mechanistic differences—see our response to Reviewer 1 on this point); and (4) under theorised dynamics of urban environments, natural selection should favour risk-sensitive learning. We therefore do not feel it is misleading to say that we mapped a full pathway from behaviour to mechanisms through to selection and adaptation. Again, as we now state in L311–313, we caution against speculating about any between-population variation, as we did not infer any meaningful behavioural or mechanistic population-level differences. And we note the Reviewer is wrong to assume an interaction between learning, dispersal, and sex requires population-level differences on the outcome scale—please see our discussion on phenotypic plasticity and inherent species trait(s) in L313–324.

L216: "indeed explain" again worded too strongly.

We have tempered our wording. Specifically, we now state in L218: “sufficiently explain”. This wording is tonally accurate with respect to the inferential value of agent-based forward simulations—please see L192–207 on this point.

L234: "reward-payoff sensitivity" might be a better term than risk-sensitivity?

Please see our earlier response to this suggestion. We note we have changed this text to state “risk-sensitive learning” rather than “reward-payoff sensitivity”, to hopefully prevent the reader from concluding only our lambda term is sensitive to rewards—a point we now include in L153–154.

L234-237: I think these points may be valuable, but come too much out of the blue. Many readers will not have a detailed knowledge of the experimental assays. It therefore also does not become clear how they measure the wrong thing, what this study does to demonstrate this, or whether a better alternative is presented herein. It almost seems like this should be a separate paper by itself.

We apologise for this lack of context. We now explicitly state in L275 that we are discussing reversal learning assays, to give all readers this knowledge. In doing so, we hope the logic of our argument is now clear: reversal learning assays do not measure behavioural flexibility, whatever that even is. The Reviewer’s suggestion of a separate paper focused on what reversal learning assays actually measure, in terms of mechanism(s), is an interesting one, and we would welcome this discussion. But any such paper should build on the points we make here.

L270-288: Somewhere here the authors have to explain how they have not found differences between populations, or that in so far as they found them, they run against the originally stated hypothesis.

We thank the Reviewer for these suggestions. In L310—313 we now state: “The lack of spatial replicates in the existing data set used herein inherently poses limitations on inference. Nevertheless, the currently available data do not show meaningful population-level behavioural or mechanistic differences in grackles’ reinforcement learning, and we should thus be cautious about speculating on between-population variation”.

L284: should be "missing" not "missed out"

We have made this change.

L290-291: It is unclear what "robust interactive links" were found. A pattern of sexbiased learning was found, which can potentially be attributed to evolutionary pressures in urban environments. An interaction e.g. between learning, dispersal, and sex can only be tentatively suggested (no differences between populations). Also "fully replicable" is a bit misleading. The analysis may be replicable, but the more relevant question of whether the findings are replicable we cannot presently answer.

We apologise for our lack of clarity. By “robust” we mean “across population”, which we now state in L333. We again note the Reviewer is wrong to assume an interaction between learning, dispersal, and sex requires population-level differences on the outcome scale— please see our discussion on phenotypic plasticity and inherent species trait(s) in L313–324. Finally, the Reviewer makes a good point about our analyses but not our findings being replicable. In L334 we now make this distinction by stating “analytically replicable”.

L306-315: I think you have a bit of a sample size issue not so much when populations are pooled but when separated. This might also factor in the fact that you do not really find differences across the populations in your analysis. When we look at the results presented in Figure 2 (and table d), we can see a trend towards males having better risk sensitivity in core (HPDI above 0) and middle populations (HPDI barely crossing 0) but the difference is very small. Especially the results on females are based on the performance of only 8 and 4 females respectively. I suggest making this clear in the manuscript.

In Bayesian statistics, there is no strict lower limit of required sample size as the inferences do not rely on asymptotic assumptions. With inferences remaining valid in principle, low sample size will of course be reflected in rather uncertain posterior estimates. We note all of our multilevel models use partial pooling on individuals (the random-effects structure), which is a regularisation technique that generally reduces the inference constraint imposed by a low sample size (see Ch. 13 in Statistical Rethinking by Richard McElreath [PDF: https://bit.ly/3RXCy8c]). We further note that, in our study preregistration (https://osf.io/v3wxb), we formally tested our reinforcement learning model for different effect sizes of sex on learning for both target parameters (phi and lambda) across populations, using a similarly modest N (edge: 10 M, 5 F; middle: 22 M, 5 F ; core: 3 M, 4 F) to our actual final N, that we anticipated to be our final N at that time. This apriori analysis shows our reinforcement learning model: (i) detects sex differences in phi values >= 0.03 and lambda values >= 1; and (ii) infers a null effect for phi values < 0.03 and lambda values < 1 i.e., very weak simulated sex differences (see Figure 4 in https://osf.io/v3wxb). Thus, both of these points together highlight how our reinforcement learning model allows us to say that across-population null results are not just due to small sample size. Nevertheless the Reviewer is not wrong to wonder whether a bigger N might change our population-level results; it might; so might muchneeded population replicates—see L310. But our Bayesian models still allow us to learn a lot from our current data, and, at present, we infer no meaningful population-level behavioural or mechanistic differences in grackles’ behaviour. To make clear the inferential sufficiency of our analytical approach, we now include some of the above points in our Statistical analyses section in L452–457. Finally, we caution against speculating on any between-population variation, as we now highlight in L311—313 of our Discussion.

Figure 2: I think the authors should rethink their usage of colour in this graph. It is not colour-blind friendly or well-readable when printed in black and white.

We used the yellow (hex code: #fde725) and green (hex code: #5ec962) colours from the viridis package. As outlined in the viridis package vignette (https://cran.rproject.org/web/packages/viridis/index.html), this colour package is “designed to improve graph readability for readers with common forms of color blindness and/or color vision deficiency. The color maps are also perceptually-uniform, both in regular form and also when converted to black-and-white for printing”.

Figure 3B: Could the authors turn around the x-axis and the colour code? It would be easier to read this way.

We appreciate that aesthetic preferences may vary. In this case, we prefer to have the numbers on the x-axis run the standard way i.e., from small to large. We note we did remove the word ‘Key’ from this Figure, in line with the Reviewer’s point about these characteristics not being totally certain.

I also had a look at the preregistration. I do think that there are parts in the preregistration that would be worth adding to the manuscript:L36-40: This is much easier to read here than in the manuscript.

We changed this text generally in the Introduction in our revision, so we hope the Reviewer will again find this easier to read.

L49-56: This is important information that I would also like to see in the manuscript.

We no longer have confidence in these findings, as our cleaning of only one part of these data revealed considerable experimenter oversight (see ‘Learning criterion’).

L176: Why did you remove the random effect study site from the model? It is not part of the model in the manuscript anymore.

The population variable *is* part of the RL_Comp_Full.stan model that we used in our manuscript to assess population differences in grackles’ reinforcement learning, the estimates from which we report in Table C and D (please note we never coded this variable as “study cite”). But rather than being specified as a random effect, in our RL_Comp_Full.stan model we index phi and lambda by population as a predictor variable, to explicitly model population-level effects. Please see our code:

https://github.com/alexisbreen/Sex-differences-in-grackles-learning/blob/main/Models/Reinforcement%20learning/RL_Comp_Full.stan

L190-228: I am wondering if the model validation should also be part of the manuscript as well, rather than just being in the preregistration?

We are not sure how the files were presented to the Reviewer for review, but our study preregistration, which includes our model validation, should be part of our manuscript as a supplementary file.